

# How can geologic decision making under uncertainty be improved?

Cristina G. Wilson[1,2], Clare E. Bond[3], and Thomas F. Shipley[4]

[1]Department of Psychology, College of Arts and Sciences, Temple University, Philadelphia, PA, USA
[2]Department of Electrical and Systems Engineering, School of Engineering and Applied Science, University of Pennsylvania, Philadelphia, PA, USA
[3]Department of Geology and Petroleum Geology, School of Geosciences, University of Aberdeen, Kings College, Aberdeen, UK
[4]Department of Psychology, College of Arts and Sciences, Temple University, Philadelphia, PA, USA

**Correspondence:** Cristina G. Wilson (cristina.wilson@temple.edu)

**Abstract.** In the geosciences, recent attention has been paid to the influence of uncertainty on expert decision making. When making decisions under conditions of uncertainty, people tend to employ heuristics (rules of thumb) based on experience, relying on their prior knowledge and beliefs to intuitively guide choice. Over 50 years of decision making research in cognitive psychology demonstrates that heuristics can lead to less-than-optimal decisions, collectively referred to as biases. For example,

the availability bias occurs when people make judgments based on what is most dominant or accessible in memory; a geoscientist who has spent the past several months studying strike-slip faults will have this terrain most readily available in her mind when interpreting new seismic data. Given the important social and commercial implications of many geoscience decisions, there is a need to develop effective interventions for removing or mitigating decision bias. In this paper, we outline the key insights from decision making research about how to reduce bias and review the literature on debiasing strategies. First, we

define an optimal decision, since improving decision making requires having a standard to work towards. Next, we discuss the cognitive mechanisms underlying decision biases and describe three biases that have been shown to influence geoscientists decision making (availability bias, framing bias, anchoring bias). Finally, we review existing debiasing strategies that have applicability in the geosciences, with special attention given to those strategies that make use of information technology and artificial intelligence (AI). We present two case studies illustrating different applications of intelligent systems for the debiasing

of geoscientific decision making, where debiased decision making is an emergent property of the coordinated and integrated processing of human-AI collaborative teams.

## 1 Introduction

"Evidently, if the investigator is to succeed in the discovery of veritable explanations of phenomena, he must be fertile in the

invention of hypotheses and ingenious in the application of tests. The practical questions for the teacher are, whether it is possible by training to improve the guessing faculty, and if so, how it is to be done. To answer these, we must give attention





to the nature of the scientific guess considered as a mental process. Like other mental processes, the framing of hypotheses is usually unconscious, but by attention it can be brought into consciousness and analyzed." – Gilbert (1886)

When G.K. Gilbert wrote about the development of a "guessing faculty" in "*The Inculcation of Scientific Method by Example*" (1886), he was one of the first to highlight the value of understanding how geoscientists resolve epistemic uncertainty during judgment and decision making. Epistemic uncertainty refers to knowledge that an individual, in principle, could have, but does not, i.e., limited information of the environment or system of study. Although epistemic uncertainty is a feature of all sciences, in the geosciences it is the standard rather than the special case (Bárdossy and Fodor, 2001; Frodeman, 1995). Geoscientists must frequently make decisions where data are incomplete, e.g. where the rock record is incomplete due to limited exposure or erosion, where isolation of processes can be difficult because multiple processes have cumulatively transformed the rocks, and where direct observation (much less experimental control) is impossible due to the large time spans of geologic processes, which leaves evidence lost or buried beneath the Earth's surface.

To understand the influence of uncertainty on decision making in the geosciences, and what the human mind adds to the problem of inducing rules from incomplete cases, recent research has followed Gilbert's advice and studied geologic uncertainty as a mental process through the lens of cognitive science. In this work, particular attention has been paid to ways that humans constrain judgment and decision making through the use of heuristics, i.e., rules of thumb. Heuristics are efficient and offer satisfactory solutions for most decisions, but they can sometimes yield less-than-optimal choices, collectively referred to as *human decision biases*. Geoscience scholars have begun to characterize the influence of such biases in geologic decision making (Alcalde et al., 2017a, b; Barclay et al., 2011; Bond et al., 2007; Polson and Curtis, 2010; Rowbotham et al., 2010; Taylor et al., 1997). For example, the interpretation of synthetic seismic images has been shown to be vulnerable to availability bias, which occurs when people make judgments based on what is most dominant or accessible in memory; participants' interpretations were positively related to their primary field of expertise in tectonic settings, i.e., an individual who indicated thrust tectonics as their primary field (and had this setting most accessible in memory) was more likely to interpret the image as thrust faults than an individual with a different expertise (Bond et al., 2007).

Characterizing the impact of decision biases such as the availability bias is important and more work is needed to determine the range of biases influencing geoscientists and their prevalence in geologic decision making. However, given the potential costs of biased decisions, there is a still greater need to develop effective interventions for removing or mitigating bias. In the words of Gilbert (opening quote), to determine "whether it is possible by training to improve the guessing faculty, and if so, how it is to be done." The development of debiasing techniques is especially important for geologic decisions that have social and commercial implications (e.g., hazard prediction, resource extraction, waste storage, water supply), but could also benefit the underpinning workflows involved in more commonplace decisions (e.g., navigation, mapping) and result in improved field practice. The cognitive literature on judgment and decision making offers valuable insights into how susceptibility to decision bias can be reduced and thus how geologic decision making under uncertainty might be improved. In this paper, we outline the key insights from judgment and decision making research about how to reduce bias and review the literature on debiasing strategies. In doing so, we seek to highlight the most promising avenues for future research on debiasing geologic decision making in the context of evolving technological advancements in geoscience practice and education.




The paper is organized as follows. First, we briefly discuss how to define an optimal decision, since "improving" geologic decision making necessitates having a clear standard to work towards. Next, we describe the origins of decision biases using a dual-process distinction that has been supported by a wealth of research in cognitive science (for review, see Evans and Stanovich (2013)). Dual-process theories of decision making posit the existence of two unique sets of processes, a set of in-
tuitive and largely automatic processes and a set of deliberative and more effortful processes. We explain how dual-process theories can account for three specific decision biases, the availability bias, framing bias, and anchoring bias. We focus on these three biases because their influence has been well-documented in the geoscience literature. Finally, we analyze existing debiasing strategies that have applicability in the geosciences. We categorize existing strategies based on whether they debias by modifying the decision maker (i.e., provide knowledge or tools that must be self-employed to debias), or debias by modi-
fying the environment (i.e., change settings or the information available in the environment where decisions occur to debias). Special attention is given to debiasing strategies that make use of information technology and artificial intelligence (AI) when modifying the decision maker or environment. We believe that these technologies offer the opportunity to overcome some of the cognitive constraints that result in biased strategies and thus hold the greatest promise of successful application in the geosciences.

## 2   Optimal decision making

What does it mean to choose optimally? The position we take in this article is that normative decision models offer a reasonable benchmark for optimal choice, and could be applied in a geoscience context. Normative models are based in economic theory and describe how people *should* make decisions: People *should* strive to maximize the expected utility of a decision, the probability that an act will lead to an outcome that is preferable to all alternatives (the principle of dominance). Also, people
*should* be internally consistent in their decision making, meaning they should assign the same utility to decision alternatives regardless of minor changes in context, such as the description and order of alternatives or the presence or absence of other alternatives (the principle of invariance).

Unfortunately, what people should do is not always what they actually do. Numerous behavioral decision studies have demonstrated that how people make decisions in the real world can systematically violate normative models (for review, see
Gilovich et al. (2002); Kahneman (2011)). These systematic departures from optimal choice are referred to as biases, and they arise (as described in the Introduction) from reliance on heuristics during decision making. For example, framing equivalent decision outcomes as positive (e.g., 60 percent chance to win) or negative (e.g., 40 percent chance to lose) has been shown to systematically alter risk preference, a violation of the invariance principle. This framing bias can drive individuals to make decisions that fail to maximize expected value (e.g., preference for a certain gain over a gamble with a probabilistically higher
value), a violation of the dominance principle.

While it is clear from past research that people do not always make decisions as they should, there is good reason to believe that people have the capacity to improve decision making to the normative standard. This is evidenced by the observations that (1) when people actively reflect on normative principles they are likely to endorse them, even if they have violated those norms





in the past, and (2) some people already adhere to normative principles in their decision making, and these individuals often have higher analytic skills (are more reflective and engaged) than those who are vulnerable to bias (for review, see Stanovich and West (2000)). Thus, in the current article we will not focus on the question of whether geoscientists *can* make optimal choices (we assume this is possible), instead we will address the question of *how* to effectively move geoscientists decision making towards a normative standard (section 4, Debiasing Strategies and Interventions). However, first we review in more detail the cognitive mechanisms by which biases arise.

## 3  Origins of decision biases

In the opening quote, Gilbert is astute in his observation that the mental processes through which decision makers handle uncertainty are frequently outside the focus of their attention. Converging evidence from psychology and neuroscience suggests there are two distinct sets of processes driving judgment and decision making (for review, see Evans and Stanovich (2013)). One set of processes is *intuitive* and places minimal demand on cognitive resources because it does not require controlled attention. The other set is *deliberative* and places greater demand on cognitive resources, but also enables uniquely human abilities such as mental simulation and cognitive decoupling during hypothetical thinking, i.e., the ability to prevent real world representations from becoming confused with representations of imaginary situations. In the judgement and decision making literature these sets of processes are typically referred to as Type 1 and Type 2 processes, respectively; however, for the purposes of this paper we will simply refer to them as *intuitive* and *deliberative*. Both sets of processes serve important functions when making decisions under uncertainty. Generally, intuitive processes are believed to be prompted rapidly and with minimal effort, providing default responses that serve as heuristics. When the heuristic responses are inappropriate and do not align with set goals, deliberative processes intervene and engage available resources for slower, more reflective reasoning (Evans and Stanovich, 2013; Kahneman and Frederick, 2002).

The interaction between intuitive and deliberative processes can be likened to that of a reporter and an editor in a newspaper room[1]. Reporters (i.e., intuitive processes) interpret the world and produce the bulk of the written work (i.e., decision making). It is the job of the editor (i.e., deliberative processes) to endorse the work of reporters, edit the work, or stop it altogether. Unfortunately, editors are often overburdened, and so stories that should be edited or stopped (because they are in some way flawed or objectionable) are instead endorsed. Similarly, deliberative processing is often overworked because it is restricted by the limited capacity of available cognitive resources – and so heuristic responses that are ill-suited for the current decision environment can be mistakenly endorsed by deliberative processes.

It is important to note that, on most occasions, heuristic responses do lead to good decisions. Intuitive processing draws from our knowledge of the world, our experiences, the skills we possess – it is what allows us to move quickly and efficiently through our environment, making decisions with relatively little effort. To truly appreciate the value of intuitive processing, consider how your behavior in the last hour would have changed if you had to *deliberately think* about every choice: you would

---

[1]1. The reporter-editor analogy, to our knowledge, was first introduced by Daniel Kahneman in 2013 during an interview with Morgan Housel for The Motley Fool, a multimedia financial advisement company.





have had to deliberate about the best form for sitting down and standing up, the amount of time spent in each position, where to put your pen, which way to position your coffee cup, the best web browser to use, and so on. Viewed in this light, we should all be thankful for the intuitive processes that allow us to make decisions quickly and effortlessly.

Yet, while intuitive processing is generally effective, there are some situations in which the heuristic responses generated by intuitive processes are inappropriate and fail to meet desired goals. In such circumstances, we need intervention from deliberative processes to behave optimally. Decision making is vulnerable to bias when (1) deliberative processes do not interrupt and override faulty heuristic responses, or (2) when the content of deliberative processing is itself flawed. Because deliberative processing is constrained by the capacity of available cognitive resources, in situations where resources are already limited (e.g., high mental effort tasks, fatigue, sleep deprivation) decision makers will be more likely to rely on heuristic responses, thus making them particularly susceptible to bias in such situations. Also, in general, humans act as cognitive misers (Böckenholt, 2012; Stanovich, 2009), meaning even when cognitive resources are available for deliberative processing, we tend to rely on less effortful intuitive processes. Thus, broadly speaking, decision making may be debiased by changing the environment, or through training change the decision maker, so default heuristic responses lead to good decisions, or so the application of deliberative processing is supported under conditions where it is otherwise unlikely to be applied (Milkman et al., 2009).

There are three decision biases that have been shown to influence geologic decision making: the availability bias, the framing bias, and the anchoring bias. All three are driven by faulty heuristic responses, which should be overridden by deliberative processes but are not. A form of anchoring bias can also be driven by flawed deliberative processing, which we will discuss after reviewing the intuitive processing causes. These three biases by no means exhaust the full range of biases that could be influencing geologic decision making under uncertainty, but they are, at present, the best-documented in the geosciences literature. For a more complete list of biases and their potential influence on geologic decision making see Baddeley et al. (2004); Bond (2015); Rowbotham et al. (2010).

## 3.1 Availability bias

This bias is driven by an availability heuristic (intuitive process), which is a tendency to make decisions based on what is most dominant or accessible in memory. To illustrate, consider the following: do air pollutants have a higher concentration outdoors or indoors? When asked to make this judgment you likely recalled (automatically and without effort) news stories related to outdoor air pollutants; maybe you visualized billowing smoke, car exhaust, or a smoggy city skyline. The ease with which examples of outdoor air pollution were drawn to mind probably led you to conclude that air pollution is more highly concentrated outdoors versus indoors. If so, you have just fallen prey to the availability bias. In fact, of those air quality studies examining both indoor and outdoor environments, over  have found higher pollutant concentrations inside (Chen and Zhao, 2011).

The availability heuristic can lead to bias because it substitutes one question (the size or frequency of a category or event) for another (the accessibility of the category or event in memory). When the ease with which something is drawn to mind is not reflective of the true size or frequency, bias occurs. There are many factors besides frequency that can make it easy to come up





with instances in memory. For example, the recency with which something has occurred (e.g., flying is perceived as more risky or dangerous immediately following a plane crash.), whether it holds personal significance (e.g., people have better attention and memory for the household tasks they complete, causing them to underestimate the contributions of their living partner), and how salient or dramatic it is (e.g., shark attacks get lots of media attention so people tend to exaggerate their frequency).

Note, then, that if a event (a) did not occur recently, (b) does not apply to you, or (c) is banal, it will lead to an impression that the event is rare (even if it is not).

In the geoscience literature, evidence of the availability bias during data interpretation has been documented in both experts (Bond et al., 2007) and students (Alcalde et al., 2017b). Bond et al. (2007) found that experts interpretations of seismic images were related to their primary field of expertise in tectonic settings, specifically, the most dominant tectonic setting in memory

was the one selected. Likewise, Alcalde et al. (2017b) found that geology students were more likely to interpret a fault in a seismic image as normal-planar as this fault type and geometry are over-represented in teaching materials, particularly those the students had encountered. After students were exposed to a greater range of fault models through a two-week training course, the range of fault interpretation type and geometry increased. The potential value of such education programs for reducing vulnerability to decision bias is discussed in Box 1, "Can better decision making be taught?".

## 3.2 Framing bias

Framing bias occurs when people respond differently to objectively equivalent judgments based on how potential outcomes are described, or framed. It is generally believed to be the result of an initial affective reaction, or affect heuristic (intuitive process), that makes certain gains particularly attractive and certain losses particularly aversive (Kahneman and Frederick, 2007). In a now classic example of framing bias, Tversky and Kahneman (1981) showed that when disease outbreak intervention programs

were framed in terms of lives saved (i.e., 200 out of 600 people will be saved OR 1/3 probability 600 people will be saved, 2/3 probability 0 people will be saved) participants preferred the sure option over the risky option, but when the same programs were framed in terms of lives lost participants preferred the risky option over the sure option[2]. This research had a huge impact in the fields of psychology and economics (as of 2018 it has been cited over 17,600 times) because it illustrated that human preference can be the product of problem description, and not actual substance. Subsequent research has shown that

frame-driven changes in risk preference are robust, occurring across a variety of populations and domains, including experts in medicine (McNeil et al., 1982), law (Garcia-Retamero and Dhami, 2013), finance (Fagley and Miller, 1997), and geoscience (Barclay et al., 2011; Taylor et al., 1997).

Early evidence of framing bias in geologic hazard risk assessment was found by Taylor et al. (1997), across two experiments. In experiment 1, participants of varying levels of expertise (high school student, undergraduate student, professional

geoscientist) were asked to make decisions regarding hazardous waste storage, flood protection, and volcano monitoring. These problems were presented in a format similar to the disease outbreak problem by Tversky and Kahneman (1981): for the waste storage problem, the positive frame described the probability of safe storage and the negative frame described the probability of

---

[2]This example represents only one manifestation of framing bias, referred to as "risky choice framing". For a complete typology of framing effects see Levin (1998).



an accidental spill; for the flood protection problem, the positive frame described the probability the protection would succeed and the negative frame described the probability it would fail; and for the volcano monitoring problem, the positive frame described the probability the volcano would remain dormant and the negative frame described the probability of an eruption. Across all scenarios, participants demonstrated evidence of frame-driven changes in risk preference. Importantly, professional geoscientists were just as vulnerable to bias as students, suggesting that even experts (who regularly make decisions that impact

public safety) can be swayed by superficial choice descriptions.

In experiment 2 (Taylor et al., 1997), high school student participants completed a variation of the volcano monitoring problem in which they played the role of a volcanologist who must interpret incoming information from three instruments to make ongoing decisions about how many people to evacuate from the area surrounding a volcano. Readings from the three instruments were correlated with probabilities of volcanic activity that were positively framed (i.e., dormant) or negatively

framed (i.e., eruption). Participants completed either a paper-pencil version or a computerized version of the task. Again, participants demonstrated frame-driven changes in risk preference, but only in the paper-pencil version; in the computerized version, participants were resistant to framing bias. In a follow-up study by Barclay et al. (2011), the same computerized volcano monitoring problem was used but instrument readings were either presented in text format (as in experiment 2; Taylor et al. (1997)) or in a novel graphical format. Barclay et al. (2011) reported similar findings of resistance to framing in the

text version, but found that presenting instrument readings graphically produced frame-driven changes in risk preference. That presentation mode (paper-pencil, computer) and presentation format (graphical, text-based) have an influence on vulnerability to framing bias demonstrates the complexity of characterizing cognitive biases and the need for evidence informed practices, which we discuss further in section 4, Debiasing Strategies and Interventions.

### 3.3    Anchoring bias

Anchoring is the result of focusing on the first available value or estimate for an unknown quantity before making a judgment or decision about that quantity. The initial value "anchors" subsequent judgments, so decisions stay close to the value considered. Bias occurs when the anchor is incorrect or arbitrary. Unfortunately, even when decision makers are fully aware that the anchor is a meaningless value, it still has a strong influence on their choice. For example, a study by Englich et al. (2006), asked expert judges to make a hypothetical prison sentence for a shoplifter (in months) after rolling a loaded die that only landed on

three or nine. Those judges who rolled a nine gave an average sentence of eight months, while those who rolled a three gave an average sentence of five months.

In science, anchors come in the form of initial hypotheses or interpretations. Vulnerability to bias can make scientists reluctant to accept alternative explanations, even in the face of disconfirming evidence. Ultimately, this can disrupt the evolution of knowledge. In the geosciences, for example, Rankey and Mitchell (2003) demonstrated that experts only made minor changes

to their initial interpretations of 3D seismic data after being given additional quality information that could aid interpretation. One expert (who did not change his interpretation at all) noted, "I did . . . not want to change any of my picks based on the additional well data - looks like I had it nailed."



A particular form of anchoring bias is called *herding* (Baddeley, 2015). Herding is group-driven behavior in which members' judgments are anchored to those of influential group members. This can be especially detrimental in science because evidence that conflicts with established consensus or opinion can be sidelined, and if the conflicting findings are successfully published, the authors risk being ostracized or punished. There are well-known historical examples: Galileo Galilei was convicted of heresy by the Catholic church for supporting the Copernican theory that the earth and planets revolve around the sun (Lindberg, 2003); Alfred Wegener's championing of plate tectonics theory was ignored, mocked, and deemed pseudoscience by his peers
for over 40 years (Vine, 1977).

Empirical evidence of herding in the geosciences was first demonstrated by Phillips (1999) who showed that experts' probability distributions of corrosion rates for nuclear waste storage containers differed depending on whether they were elicited independently or in groups. Experts made different prior assumptions resulting in probability distributions that were initially radically different. Inter-expert discussion resulted in some convergence of probability distributions but was also accompanied
by an increase in the variance of each independent distribution, i.e., experts increased the spread of their initial distributions to encompass the spread of the consensus distribution. In a similar study, Polson and Curtis (2010) showed that experts' estimated probability distributions for the presence of a fault were vulnerable to herding; the group of experts moved towards a single member's opinion, such that the consensus distribution was primarily a reflection of the views of one individual.

As mentioned in section 3, Origins of Decision Biases, anchoring is typically driven by a faulty heuristic response, which
should be overridden by deliberative processing, but is not. When given an anchor, intuitive processes immediately construct a world where the anchor is true by activating compatible memories. This, in turn, primes decision makers to notice information that is consistent with the anchor and ignore or discount information this is inconsistent. In the Rankey and Mitchell (2003) study, the participant who "had it nailed" was demonstrating an anchoring bias driven by faulty intuitive processing; the participant was primed to interpret new data as consistent with his initial hypothesis and ignore disconfirming data.

Alternatively, anchoring can be driven by flawed deliberative processing, which occurs when decision makers fail to adequately adjust from the anchor. We know this is a failure of deliberative processing because people are more prone to insufficient adjustment when their mental resources are depleted, e.g., when their attention is loaded or when consuming alcohol (Epley and Gilovich, 2006). Research shows people tend to only adjust estimates to the first plausible value and are generally unwilling to search for more accurate estimates (Epley and Gilovich, 2006). This may explain why participants in the Phillips (1999) study,
when given the opportunity to adjust their initial probability distributions following group discussion, primarily increased their distribution range to encompass the spread of the consensus distribution. The participants made the simplest adjustment that was still a plausible reflection of the true distribution.

## 4  Debiasing strategies

Cognitive research on heuristics and biases, beginning in the 1970s and continuing today, has demonstrated the pervasive and
robust influence of an ever-increasing list of decision biases. In comparison, our understanding of how to debias decision making is limited. This is in part due to the relatively unappealing nature of debiasing research; "it is more newsworthy to show



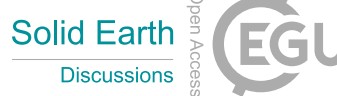

that something is broken than to show how to fix it" (Larrick, 2004). It is also likely that researchers have been dissuaded from pursuing debiasing research because early studies found biases were generally robust in the face of commonsense corrective measures, including providing feedback or incentives (Camerer and Hogarth, 1999; Fischhoff, 1982), holding people accountable for their decisions (Lerner and Tetlock, 1999), and offering warnings about the possibility of bias (Fischhoff, 1982). While there is a strong need for additional research on debiasing, a number of successful strategies have been discovered (for review, see Larrick (2004); Milkman et al. (2009); Soll et al. (2016). These existing strategies can be categorized into one of two

approaches: (1) debiasing by modifying the decision maker or (2) debiasing by modifying the environment (Soll et al., 2016). In the remaining section we consider the pros and cons of the two approaches and discuss how each can be best used in geoscience education and industry to minimize bias. We focus our discussion on debiasing strategies that make use of technology when modifying both the decision maker and environment. Access to technology has already transformed decision making in geoscience and other scientific fields, and continued advancements in technology have the potential to further aid decision

making under uncertainty.

## 4.1 Modifying the decision maker

Debiasing strategies that modify the decision maker provide knowledge and tools that must be self-employed to overcome bias. This includes cognitive strategies to shift perception of a problem (i.e, consider the opposite), the use of computational models to assist judgment, and education on statistical rules and normative principles (see Box 1). All of the above are likely familiar to

geoscience scholars, though perhaps not framed as decision aids. In fact, it can be argued that the cognitive debiasing strategy of "consider the opposite" has been a feature of geoscience for over a century. In his 1890 publication "*The Method of Multiple Working Hypotheses*", T.C. Chamberlin advocates that geoscientists should generate multiple plausible alternatives to explain the occurence of geologic phenomena (essentially a "consider the opposite" strategy). Chamberlin is clear in stating the value of this method in guarding against bias:

"The effort is to bring up into view every rational explanation of new phenomena, and to develop every tenable hypothesis respecting their cause and history. The investigator thus becomes the parent of a family of hypotheses: and, by his parental relation to all, he is forbidden to fasten his affections unduly upon any one. In the nature of the case, the danger that springs from affection is counteracted... Having thus neutralized the partialities of his emotional nature, he proceeds with a certain natural and enforced erectness of mental attitude to the investigation."

Cognitive research has supported Chamberlin's assertions that generating multiple alternatives to a decision problem can be an effective debiasing strategy, particularly when bias is driven by a tendency to rely on small and unrepresentative samples of information, as with anchoring bias (e.g., Mussweiler et al., 2000), overconfidence (e.g., Koriat et al., 1980), and hindsight bias (e.g., Sanna and Schwarz, 2006). In modern geoscience practice, as noted by Bond (2015), the method of multiple working hypotheses is not consistent with the culture of science in which advocacy for a single model is rewarded. However, there is

recognition of the value of promoting a workflow that increases consideration of permissible interpretations in the geosciences (Bond et al., 2008; Macrae et al., 2016), as well as other sciences faced with complexity and high uncertainty (Elliot and Brook,





2007). While geoscience educators recognize the centrality of the principle of multiple working hypotheses to geoscience practice there is, to our knowledge, no accepted pedagogy for supporting practice in the skill.

The debiasing strategy of applying computational models is a more recent fixture of geoscience, and its increased prominence is at least partially owed to the increased availability of user-friendly modeling software. In sustainability and resource management research in particular, many modeling and simulation software tools have been created to aid judgment (for review see Argent (2004); Argent and Houghton (2001); Rizzoli and Young (1997). Computational models systematize the weights placed on various decision inputs (in lieu of relying on expert experience), enabling better forecasting of outcomes. In situa-

tions where expert experience is critical to predicting outcomes, that experience can become a decision input in the model. Past research has shown that such computational models outperform expert judgment in a variety of domains (Dawes et al., 1989), and even the simplest linear models that equally weight all relevant decision inputs (not taking into account historical data on inputs) can outperform expert judgments (Dawes and Corrigan, 1974). In the geosciences, recent research investigating the impact of model use on human judgment about resource management found that management outcomes were superior when

student participants relied on models rather than their own experience (Holden and Ellner, 2016). Also, model use has been shown to improve sustainability policy decisions – in a role play simulation, model users evidenced better outcomes (e.g., low change in temperature, high access to electricity, and high global economy) than participants who did not use models (Czaika and Selin, 2017).

For decision makers to use computational models or employ cognitive techniques like "consider the opposite" successfully

requires (at minimum) that deliberative processing resources be available for suspending and correcting decisions. These debiasing strategies operate by supporting the application of deliberative processing under conditions where it is likely to not be applied, encouraging decision makers to shift themselves from intuitive to deliberative processing. For example, when decision makers "consider the opposite" it encourages deliberative analysis and suspends reliance on intuitions that distort the representation of information (e.g., intuition to rely on what is most dominant or accessible in memory). Recall from section 3, Origins

of Decision Biases, that we have a limited capacity of deliberative resources to draw from, i.e., our "editor" can be overworked. Therefore, these debiasing strategies will have a low probability of success in situations where deliberative processing resources are reduced (e.g., high mental effort tasks, states of fatigue or sleep deprivation). Also, individual differences in thinking style and quantitative ability (i.e., numeracy) impact the effectiveness of deliberative processing debiasing strategies; people who are reflective and have high numeracy are more likely to suspend heuristic responses and invoke deliberative resources to do

necessary additional thinking (Frederick, 2005). However, even under ideal conditions, where deliberative resources are not constrained and the decision maker is prone to reflective thinking, there is no guarantee of debiasing success – and herein lies the problem with self-employed debiasing strategies, they may require too much of the decision maker. Successful implementation requires that the decision maker be able to recognize the need to apply a strategy, have the motivation and the required deliberative resources to do so, select the appropriate strategy and apply it correctly. A mistake or failure at any step of this

process could result in (at best) continued vulnerability to decision bias or (at worse) an increase in bias.

Consider, for example, the application of computational models in sustainability and resource management research. Although there are many modeling and simulation software tools available for forecasting climate outcomes (for review see





Argent (2004); Argent and Houghton (2001); Rizzoli and Young (1997)), there is concern amongst geoscience scholars that decision makers are not using models as often as expected, or correctly (see also Box 1 for discussion of concerns about the

level of quantitative education in the geosciences). Both Edwards et al. (2010) and Oxley et al. (2004) found poor receptivity to and "low uptake" of modeling software tools amongst targeted end users within EU funded research projects. The authors of these studies argue that low uptake resulted from bad communication between tool developers and end users. Thus, despite computational models being available for use in forecasting climate outcomes, some experts are not sufficiently motivated to apply them or are unconfident in their ability to select the appropriate model and apply it correctly, instead relying on their

experience and intuition as a substitute for formal analysis.

Determining methods for facilitating the adoption of self-employed debiasing strategies is a critical issue for debiasing research both generally and in the geosciences. Some of the reluctance to use computational models in the geosciences can be solved by improving the design and user-interface of modeling and simulation software. To this end, McIntosh and colleagues have outlined design principles (McIntosh et al., 2005) and best practices (McIntosh et al., 2008) for the development of

computer-based models, with the goal of improving the usefulness and usability of modeling tools in the geosciences. However, even with improved tool design, decision makers may continue to resist using computational models and other self-employed debiasing strategies. In the words of debiasing researcher Richard Larrick 2004:

"[Decision makers] do not want to be told that they have been 'doing it wrong' for all these years. They do not want to relinquish control over a decision process. And, perhaps most importantly, they fail to understand the benefits of many debiasing

techniques relative to their own abilities, not just because they are overconfident, but because the techniques themselves are alien and complex, and the benefits are noisy, delayed, or small."

In sum, self-employed debiasing strategies carry a high risk of being used inappropriately by decision makers (i.e., used incorrectly or not at all), and for this reason we believe that such strategies *alone* do not offer the most promise for successful application in the geosciences. Instead, we advocate that debiasing strategies (including "consider the opposite" and using

computational models) be supported by modifying the decision environment such that (1) people are "nudged" towards the optimal choice strategy or (2) the environment becomes a good fit for the strategy people naturally apply, thereby relieving decision makers of the impetus for debiasing.

### 4.1.1   Box 1. Can better decision making be taught?

There is good evidence that decision making can be improved by teaching people statistical rules and normative principles.

Research in this area examines the influence of teaching across formal higher education, disciplinary-specific training, single courses, and brief laboratory sessions. For example, Doctoral-level scientists, with extensive statistical training, are better than psychology graduate students, with two to three courses in statistics, at applying statistical rules to avoid drawing inferences from small samples – but graduate students do better than undergraduate students with only one statistics course (Fong et al., 1986). Also, economics professors are more likely than biology and humanities professors to use normative principles

in everyday decision making, such as ignoring a sunk cost by leaving a mediocre play early in the performance (Larrick et al., 1993), and college students can be taught how to apply normative principles in laboratory sessions lasting less than an



hour (Larrick et al., 1990). From this research we infer that better decision making in the geosciences can also be taught by a substantive statistics curriculum.

Although modern geoscience researchers have embraced statistical methods, and quantitative skills are fundamental to the evaluation and investigation of geologic processes, there is not a strong history of statistics curriculum in undergraduate and graduate geoscience courses (Manduca et al., 2008). However, in the past 20 years there has been increased interest in identifying quantitative skills that students need to succeed in the field and developing strategies for teaching those skills; see Kempler and Gross (2018) for a recent example. As a result, statistical training is now a more common feature of geoscience education. In future research, it would be interesting to know if education on statistical rules in the geosciences mitigates some biases (e.g., sunk costs), and how any improvements in decision making compare to those achieved through similar statistical education in other sciences. Also, it would be worthwhile to determine the reliability and duration of decision improvement following statistics education: do geoscientists with statistical education always avoid drawing inferences from small samples, or just occasionally, and how long after education has concluded do improvements endure?

Beyond education on statistical rules and normative principles, in some fields it is common that students receive additional decision-focused curriculum. For example, in fields such as business and medicine, where there has been longer recognition of the influence of decision bias, students are taught how experts resolve uncertainty in decision making, and the biases that occur when experts rely on heuristics that are not well suited to the choice environment. To our knowledge, most of this decision-focused curriculum is descriptive, that is, it teaches facts about biases (including a taxonomy of biases) and how they distort reasoning, but does not address strategies for overcoming bias. To date, the effectiveness of courses with decision curriculum aimed at reducing vulnerability to bias is unknown.

We feel that geoscience education and industry would benefit from the adoption of similar decision-focused curriculum, since being aware of the existence and possibility of decision bias is the first necessary step to reducing vulnerability. Already uncertainty training has been incorporated into many major oil company training portfolios and is offered by training consultants to the geoscience industry. Yet, past research would suggest that simply being aware of the possibility of bias is not enough to reduce susceptibility by any substantial margin (Fischhoff, 1982). A potentially worthwhile addition, then, to decision curriculums in geoscience and other fields would be education on choice architecture and "nudging" (see section 4.2, modifying the environment), i.e., teaching students how to structure and engage with their environment to promote good judgment and decision making. This would include instruction on how and why biases occur and debiasing strategies to mitigate them, but also practice with choice infrastructure creation so decision makers are not required to self-employ strategies to rise above their ingrained and subtle biases. How best to teach these courses and what kind of content, guidance, and practice to offer is an important question for future education research (for discussion on the possible virtues of "nudge" education, see Beaulac and Kenyon (2014); for discussion of how to achieve such institutional education changes see Henderson et al. (2015)

## 4.2 Modifying the environment

As stated above, debiasing techniques that modify the environment alter the settings where decisions occur – this can "nudge" people towards the optimal choice strategy (e.g., prompts to induce reflection and deliberation), or make the environment a




better fit for the strategy people naturally apply (e.g., status-quo bias pushes people to stick with a default response option over selecting a new option, so making the default a desirable outcome will maximize decision making). This approach to debiasing is sometimes referred to as *choice architecture*, making the individual or entity responsible for organizing the environment in which people make decisions the *choice architect* (Thaler and Sunstein, 2008). It is the role of the choice architect, as put forward by Thaler and Sunstein (2008) in their popular press book "*Nudge*", to influence people's decision making such that their well-being (and the well-being of others) is maximized, without restricting the freedom to choose. Importantly, there is no such thing as neutral choice architecture; the way the environment is setup will guide decision making, regardless of

whether the setup was intentional on the part of the architect, e.g., descriptions of risk will be framed in terms of gains or losses, a wise architect chooses the framing that will maximize well-being.The advantage of debiasing techniques that modify the environment, over those that modify the decision maker, is that it is the choice architect and not the decision maker who is accountable for debiasing (unless, of course, the architect and the decision maker are the same person). Conscious choice architecture is a naturally deliberative process – potential mechanisms of bias must be considered and used to design

nudges, user responses to these nudges must also be considered, including factors unrelated to the nudge that may influence responses. Therefore, techniques that modify the environment tend to be more successful in reducing vulnerability to bias and improving decision making, and this has been evidenced in varied domains, e.g., improving rates of organ donation (Johnson and Goldstein, 2003), increasing employee's retirement savings (Madrian and Shea, 2001), and encouraging healthier eating habits (Downs et al., 2009).

Choice architecture debiasing techniques have been adopted in the sciences in the form of imposed workflow practices and structured expert elicitation exercises. The latter may be more familiar to geoscientists given there is a long history of using cumulative expert judgments in geologic research when data are insufficient (e.g., Cooke and McDonald, 1986; Hemming et al., 2018; Wood and Curtis, 2004). Expert elicitation research demonstrates that structured methods can be employed to enforce consideration of group ideas and opinions, such that the vulnerability to overconfidence and other biases is reduced (Wood and

Curtis, 2004; Polson and Curtis, 2010). The use of imposed workflows, in comparison, is a newer feature of scientific practice. In the social and life sciences, a research reproducibility crisis has led many academic gatekeepers to advocate for the use of workflows such as study preregistration (i.e., a description of study methods, materials, and analyses published prior to data collection) and open sharing of data and study materials. In geoscience research, use of workflows is also increasingly encouraged. For example, Gil et al. (2016) propose that the "Geoscience Paper of the Future" should make data and software

reusable and explicitly describe higher-level computational workflows.

One way the value of choice architecture debiasing in the sciences manifests is through improved efficiency and effectiveness of an ongoing decision process – how much data is enough to reach a conclusion about my interpretation(s) and how confident am I in that conclusion? This decision process can be visualized by plotting the relationship between conclusion certainty and data; where data is generally defined as the quality and quantity of information known. Notably, in most sciences, amount

of data (as defined above) is directly proportional to time, resources, and funds. As data accumulates, conclusion certainty increases until some threshold of confidence is reached, at which point the scientific decision maker makes a conclusion about her interpretation(s). We define this threshold of confidence as a geologists individual perception of being "certain enough"




in their interpretation, and hence the threshold can differ dramatically between individuals (what seems certain to one will not seem certain to all) and be shaped by research context (what counts as "certain enough" in one research field will not count in all research fields). In the ideal case, where data is homogeneous, collected in an unbiased manner, and is consistent with a theory. there is a positive linear relationship between data and conclusion certainty, with greater slopes indicating greater decision efficiency, i.e., faster ascension to the decision makers threshold of confidence (see Fig. 1). However, as every researcher knows, this ideal case is rare. More often, as data accumulates, researchers experience upward and downward shifts in conclusion certainty. Decision biases can impact how confidence shifts with incoming data, and where the threshold of confidence is set.

Consider the following study by Macrae et al. (2016) on geologic workflows as an example. Macrae et al. (2016) gave geoscientists a 2D seismic reflection image and asked them to provide an interpretation within a limited time frame. Half the geoscientist participants were asked to undertake a specific workflow in which they received instruction to explicitly consider geological evolution through writing or drawing (a choice architecture debiasing technique), and the other half received no workflow. The results from Macrae et al. (2016) reveal that geoscientists nudged to consider geological evolution had higher quality interpretations than those who received no nudge. Because real seismic data was used, the correct interpretation was unknown, but interpretations were deemed high quality if they were consistent with the interpretations of at least one of five leading experts. Some participants may have been overconfident in their interpretation (as suggested by Macrae et al. (2016)), which implies that their threshold of confidence was lower thus increasing the likelihood of accepting an erroneous interpretation. The workflow nudge could have mitigated overconfidence by testing the interpretation (during or after creation) to determine whether the final interpreted geometry could evolve in a geologically reasonable manner. Figure 2 shows the decision process of two hypothetical individuals, Participant 1 and Participant 2. For those interpretations in which the evolution was not feasible (Participant 1, Interpretation A), the workflow nudge would force the participant to consider modifications (1B), or alternative interpretations (1C), thereby reducing overconfidence. For those interpretations in which the evolution was reasonable, certainty in interpretation would likely increase to the threshold of confidence (Participant 2, Interpretation A).

We can also use the data-certainty plot to visualize how the choice architecture practice of expert elicitation influences decision making and vulnerability to herding (anchoring) in the aforementioned study by Polson and Curtis (2010). Polson and Curtis (2010) asked four expert geoscientists to assess the probability that a fault existed. Figure 3 shows the decision process of of the four participants. Note that the x-axis of Fig. 3 is represented as time rather than data, as previously stated we view the quality or quantity of known information (data) as related to time and in this example time is the more coherent descriptor. After making their initial assessment, participants were alerted to common biases in expert judgment and allowed to modify their assessment. If as a result of the warning the participant felt they had made a mistake or their initial probability was somewhat biased, they could have modified their interpretation (Participant 1 and 3) or experienced a reduction in certainty (Participant 4). Alternatively, if the participant perceived their initial analysis to be free of bias – either because it truly was, or because the warning was not sufficient for resolving bias – then they would likely stick with their initial probability distribution and potentially experience an increase in conclusion certainty (Participant 2). Following the bias warning and any changes to



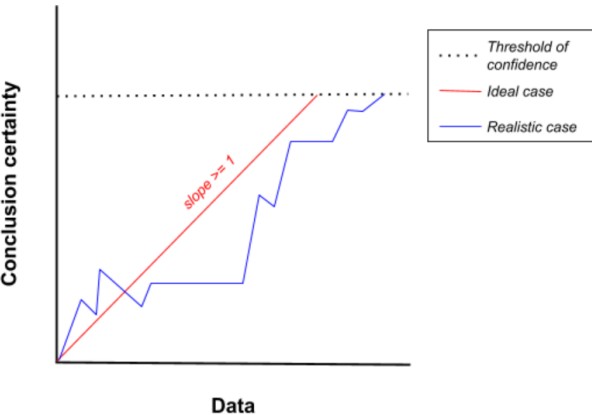

**Figure 1.** The relationship between data and conclusion certainty in the scientific decision making process. In the ideal case, increasing data is accompanied by increasing certainty, with slopes greater than or equal to one indicating an efficient decision process. In reality, increasing data is more often accompanied by upward and downward shifts in certainty. A scientific conclusion is reached once a decision maker reaches their personal subjective threshold of confidence and feels "certain enough" in their interpretation.

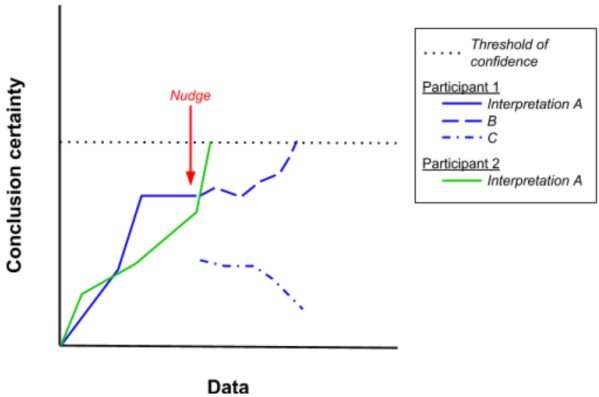

**Figure 2.** The influence of a geologic workflow nudge on conclusion certainty in a study by Macrae et al. (2016). Participant 1 and 2 both experience shifting uncertainty in their initial interpretations as data is accumulated prior to the nudge onset. After being nudged to consider the geological evolution of their interpretations, Participant 2 finds a geologically reasonable evolution for his or her interpretation, certainty in the interpretation increases to their threshold of confidence and a conclusion is made. Participant 1 cannot construct a geologically reasonable evolution of their initial interpretation, prompting consideration of modifications (1B) or alternative interpretations (1C).





the initial interpretation, experts shared their probability distributions and explained their reasoning to the group, then were asked to reach a group consensus.

The results from Polson and Curtis (2010) showed that the consensus interpretation did not reflect the opinions of all par-
ticipants in the group, instead the group moved towards one expert's opinion (Participant 1), herding around and anchoring to it (see grey zone, Fig. 3). Thus, although the bias warning may have been an effective debiasing strategy for the individual probabilities at the start of the elicitation, the experimental evidence suggests that it had minimal impact later in the elicitation process when coming to the group consensus. As discussed in section 4.1. Modifying the decision maker, there are many reasons why a simple warning may be ineffective for debiasing – it requires that the interpreter be able to recognize his or her own
bias and have the motivation and available cognitive resources to apply a strategy to combat bias – and in group situations where socially constructed hierarchies exist, bias warnings may be particularly ineffective. A potentially better debiasing strategy in this instance would have been to weight experts opinions and present this information in the form of a digital nudge; e.g., this is what the consensus probability distribution would be if a mean of the experts was calculated and it does not accommodate the range of individual expert opinions. Expert elicitations focusing on geological problems (e.g., Polson and Curtis, 2010; Randle
et al., 2019) show that much of their value is in better understanding decision making workflows, and where uncertainties and bias arise, which can be used to inform nudge design.

As the above examples illustrate, the potential for choice architecture to aid decision making in geoscience and other scientific fields is significant. Yet, choice architecture debiasing is not infallible to human error. For example, research on expert elicitation practices in the geosciences has shown that erroneous predictions about geologic events are made when using sub-
jective methods for selecting experts (Shanteau et al., 2002), and when judgments are not aggregated appropriately (Lorenz et al., 2011; Randle et al., 2019). Also, it is worth noting that Macrae et al. (2016) found over 80 percent of participants in the no-workflow group reported that they had considered geological evolution of their interpretation – *these individuals thought they were using a normative workflow*, but did not apply it effectively, either because their deliberative processing resources were overburdened from the demands of the interpretation task ("busy editor") or because of a general tendency towards cognitive
miserliness ("lazy editor"). We believe recent innovations in AI and information technology offer the opportunity to overcome the cognitive constraints that lead to both biased choice and trouble in effectively self-employing debiasing strategies. Specifically, we advocate for coordinated and integrated decision making in humans and intelligent systems, where debiased decisions are an emergent property of the human-AI collaborative team. This approach has been referred to as "captology" standing for "computers as persuasive technology" (Fogg, 2003), and, more recently, "digital nudging" (Mirsch et al., 2017; Weinmann et
al., 2016).

This paper is not the first to call attention to the utility of intelligent systems for geoscientific research. Use of intelligent technologies in geoscience research is increasingly common, e.g., mobile robotic platforms (Qian et al., 2017) and machine learning algorithms (Karpatne et al., 2018), and recent articles in *GSA Today* (Spino, 2019) and *Communications of the ACM* (Gil et al., 2018) both outline promising opportunities for intelligent systems to address research challenges in the geosciences.
Most of this existing work implicitly takes the view that humans and intelligent systems have separate but complementary functions in geologic decision making, Shipley and Tikoff (e.g., 2018). Here we present a different view, namely, that geologic





decision making can be enhanced when humans and intelligent systems work in collaboration, with a shared understanding of the task goal, relevant contextual features, and existing scientific knowledge. To illustrate the value of this digital nudging approach in geoscience research, we discuss two case studies that represent different applications of intelligent systems for the geosciences that are presently in practice: the first case study addresses the use of unmanned aerial vehicles (UAVs or "drones") to collect new field data, and the second addresses the use of software for geologic interpretation of seismic image data. For each case study, we first describe how the intelligent system is currently being used to aid geologic research, and how this application of technology has improved, or could improve, upon pre-technological research methods. Then, we describe how digital nudging can be incorporated into intelligent systems and illustrate the scientific value of nudging using the data-conclusion certainty plot (similar to Fig. 2 and 3 above).

### 4.2.1 Case Study 1. Optimizing field data collection with UAVs

Data about the Earth system is acquired through a broad range of methods. The advent of better mobile robot platforms has allowed for the deployment of robots by ground, sea, and air to collect field data at a high spatial and temporal resolution. Here, we focus on the use of aerial robots (semi-autonomous or autonomous UAVs) for data collection, but the conclusions we draw are likely applicable to other mobile robot platforms (i.e., underwater autonomous vehicles, ground robots). At present, UAVs are mainly used in geoscientific research for reconnaissance and 2D mapping. UAVs improve the efficiency of traditional "boots on the ground" reconnaissance and mapping procedures by providing access to data in large or inaccessible areas quickly. Typically data comes in the form of high resolution 2D imagery (available with commercial non-autonomous UAVs), but because UAVs have the flexibility to carry different sensors and devices, they can be customized to collect other forms of data, including hyperspectral images (Näsi et al., 2015), LiDAR (Wang et al., 2017), air quality measurement (Villa et al., 2016), and subsurface water quality measurements (Koparan et al., 2018). See Jiménez López and Mulero-Pázmány (2019) for additional sensors and devices that can be coupled with UAVs.

Once collected, UAV images can be used to construct 3D virtual outcrop models with Structure from Motion photogrammetric software, which determines camera position and orientation automatically to recreate geometry without the need for visible ground control (although as orientation and scale are important factors in much geological research ground control points are often used). These 3D models can then be interpreted and analyzed as subsurface analogues for collection of feature information out of the field (for review of applications, see Bemis et al. (2014). Recent research found a virtual outcrop model generated with UAV based Structure from Motion more accurately represented elements of a complex 3D outcrop than a model generated with ground based LiDAR, due to acquisition occlusion (Cawood et al., 2017). Thus, in addition to making field reconnaissance and mapping more efficient, UAVs can be used to produce large 3D digital datasets with up to centimetre-scale resolution that will aid future field and laboratory research.

Currently, the majority of geoscience research with UAVs is non-autonomous, i.e., user-controlled. Efforts have been made to automate interpretation of geological data from UAV imagery or 3D reconstruction with some success (Thiele et al., 2017; Vasuki et al., 2014, 2017), and the application of image analysis and machine learning techniques continue to be developed (Zhang et al., 2018). In reconnaissance and geologic mapping, the decision of where to go and how to fly there is made





by the expert – either the expert fly's the UAV and makes navigation decisions in-situ or they pre-set a flight path for the UAV to follow semi-autonomously Koparan et al. (e.g., 2018); Ore et al. (e.g., 2015). However, a UAV that is capable of attending to measurements in real time and reacting to local features of measurement data could navigate autonomously to collect observations where they are most needed. In this case study, we describe how automated UAV navigation can nudge geoscientists to be more efficient when making decisions regarding reconnaissance and mapping.

In our hypothetical example, a UAV surveys a large bedding surface with the aim of identifying fractures to define the orientations of fracture sets. The bedding surface exposure is large, but split into difficult to access exposure, e.g., due to cliff-sections or vegetation (see Column A, Fig. 4). A birds-eye view afforded by the UAV improves the ability to observe fractures, which would otherwise require time-costly on-foot reconnaissance to different outcrops of the bedding surface. Note that in our hypothetical example we assume that fracture information is obtained only when the flight path crosses fractures (e.g., Column B, blue flight path), thereby representing a high level reconnaissance rather than a flight path in which overlapping imagery is collected, e.g. to create a Structure from Motion virtual model. When the UAV flight path is user-controlled, the decision of where and how to fly is unlikely to be optimal: users could be distracted by irrelevant information in UAV view, and are likely biased towards exploring certain features and ignoring others

Column C on Fig. 4 shows a UAV flight (purple) that is semi-automated to follow a pre-set path. For this hypothetical example, a reasonable pre-set flight path (and the one we assume most experts would take) would be to scan forward and backward across the area of interest, akin to a lawn-mower. With this approach, the appearance is that no areas will be missed, the area is equally covered and there is no risk of re-sampling, but the flight path will not be optimal to collect the data of interest – time will be wasted scanning areas that have little data value. On the certainty plot in Column C this is visualized as long delays between detection of unique fracture orientations (a, b, c), resulting in a step-like pattern – dramatic decreases in certainty when a new orientation is observed during navigation, followed by periods of slowly increasing certainty as information is observed that is consistent with previous features, or irrelevant (green bars representing time over woodland). In this instance the user's threshold of certainty is reached after a longer time period than in the user driven scenario (Column B), but the full range of fracture orientations is determined (see rose diagrams, Column C).

The most efficient solution is for the UAV to move autonomously to areas with high data value by attending and reacting to measurement data in real time, e.g., skipping areas that are poorly exposed or homogenous, slowing and flying multiple angles in areas that have a high frequency of important features (as defined by the user). This is visualized in Column D (red flight path), where the UAV detects the first fracture orientation (a) and then recommends the user update the flight path to move orthogonal to the orientation to ensure it is representative and to optimize continued sampling. When a new orientation is detected (b) the UAV recommends updating the flight path again to optimize collection of both orientations (i.e., horizontal flight path). In the updated horizontal flight path, the UAV moves efficiently over exposure that features already detected orientations (a, b), which leads to quicker detection of new orientations (c). The rose diagrams in Column D show that by time T2 all three fracture sets have been identified. In fact by time T3 fractures oriented NE-SW are being oversampled; the same fracture is crossed more than once by the UAV flight path, not an issue here as we are only interested in constraining the number of orientations of fracture sets, but would need to be taken into consideration if the user wanted both orientation





and relative intensity. The accompanying certainty plot in Column D shows that this autonomous flight path results in a more efficient scientific decision making process, i.e., strong positive relationship between conclusion certainty and time and quick ascension to threshold of confidence.

We believe this case study is a useful exemplar. The goal was to detect all fracture orientations and determine the optimal location for sampling heterogeneous orientations. A UAV that possesses some representation of this goal can use multiple fracture orientation angles to rapidly calculate a flight path that will optimize continued sampling to confirm a fracture set or allow for more rapid detection of new orientations – and note that this type of calculation is a task in which a computer is likely to excel relative to the human mind. By offloading the task of navigation geometry to a UAV, the human expert free's up their cognitive resources for more important and difficult tasks, such as the real-time interpretation of surface features from UAV imagery. Were the goal of UAV reconnaissance to collect data on fracture length, orientation and intensity, the programming of the UAV and the human interaction would be different. In this manner, we view the robot and human as a collaborative team, where better decision making is a property of the coordination of both agents and a mutual understanding of the task goal and each other's strengths and weaknesses (Shipley and Tikoff, 2018). For example, critical to the success of our example would be for experts to understand how autonomous flight paths are being calculated and the conditions in which they will optimize data collection; an expert not privy to this information may mis-trust well calibrated path suggestions or over-trust path suggestions that are inconsistent with their goals. Also critical to the success of our example is that experts retain the ability to ignore autonomous path recommendations if their expertise leads them to favor an alternative path. One of the challenges in geosciences, and perhaps all sciences, is that AI systems focus only on the constrained problem, and (unlike humans) are not open to the frisson of exploring other questions enroute to the answer. Therefore, it is important that AI systems do not restrict users autonomy to override recommendations, thereby barring the exploration of ideas through too narrow data collection.

### 4.2.2   Case Study 2. Fault interpretations in 3D seismic image data

Understanding of the geometries of sub-surface geology is dominated by interpretations of seismic image data, and these interpretations serve a critical role in important tasks like resource exploration and geohazard assessment. 3D seismic image volumes are analyzed as sequences of 2D slices. Manual interpretation involves visually analyzing a 2D image, identifying important patterns (e.g., faulted horizons, salt domes, gas chimneys) and labeling those patterns with distinct marks or colors; then, keeping this information in mind while generating expectations about the contents of the next 2D image. Given the magnitude and complexity of this task, there has been a strong and continued interest in developing semi-autonomous and autonomous digital tools to make seismic interpretation more efficient and accurate (Araya-Polo et al., 2017; Di, 2018; Farrokhnia et al., 2018). For example, automated horizon tracking ("ant" or "auto-tracking) tools correlate horizon amplitudes to create surfaces. Where amplitude signals weaken (e.g. when noise to signal ratio is high) the software may make erroneous correlations resulting in "spikes", and leave "gaps" where amplitude data is lost (e.g. where it is displaced across faults). In a normal workflow the interpreter would "tidy" the automatically generated surface checking for spikes and gaps, making decisions on whether each spike and gap was geologically reasonable and manually modifying (or not) the surface based on the decision made.





Consider the example in Fig. 5, Image A, in which the auto-correlation tool has created spikes in the horizon interpretation; this highlights to the interpreter potential errors, although this nudge has not been deliberatively coded into the software. The interpreter can edit the horizon based on the horizon picks above and below to create a geologically reasonable interpretation of horizon off-sets across the fault (Image B). Now consider Image C, here an erroneous interpretation has been made, the horizon off-sets across the fault are not geologically reasonable. Other information could be employed here to flag to the interpreter that there may be an error; for example, by using horizon displacement information along the fault (see plot C1), although in this case the increase in displacement for the blue horizon may appear sharp but reasonable, i.e., maximum displacement

near the centre of the fault as mapped. An alternative is to consider sediment thickness difference between the hanging wall and the footwall. See plot C2, showing changes in the orthogonal distance between horizons across the fault and with depth, i.e., distance along the fault from X. In this plot a thickness increase between the foot wall and the hanging wall is seen for the orange horizon. The reason for this could be syn-faulting sedimentation of the orange horizon and would be a flag to the interpreter to check for evidence of syn sedimentation (e.g. wedge shaped orange sediments in the hanging wall). C2 also

shows a difference in sediment thickness between the foot wall and the hanging wall for the blue horizon, but this time the pattern is reversed (with a greater sediment thickness in the foot wall than the hanging wall) which should provide a significant nudge to the interpreter that something is likely wrong with the interpretation, or it is a strike-slip fault.

Although the fault-offset errors discussed above (and shown in Fig. 5) may appear obvious, in our experience, checks for geologically permissible interpretations are typically made by users after a complete interpretation has been finished rather

than during the process. This can lead to significant efficiency costs in the re-editing of interpretations. Still, working in 2D is relatively simple and most errors would likely be identified quickly by experts on viewing the 2D interpretations. In contrast, holding in mind and working with 3D information requires greater cognitive processing resources and is difficult to visualise and interpret even with modern advances in virtual reality and 3D rendering (these are some of the reasons why 3D seismic image data is generally interpreted in 2D). In this case study we consider how 3D information could be used with digital

nudge technology to inform fault interpretations in a 3D seismic image volume. Simple normal fault patterns show a bull's-eye pattern of greatest displacement in the centre of an isolated fault, decreasing towards the fault-tip (see Image A, Fig. 6). Consider interpreting 2D seismic image lines across the fault starting at in-line A (Image A) and working towards in-line F: with each subsequent line the displacement of horizons across the fault should increase and then decrease, although this pattern will not be known until the interpretation is completed. Holding this information on displacements for individual faults

between in-line interpretations in complicated seismic image data (e.g. with multiple faults per seismic section, Image B, Fig. 6) is incredibly challenging even for the well-practiced expert. Here we imagine a digital nudge that alerts users to discrepancies in fault displacement patterns, thereby relieving some of the cognitive burden of 3D interpretation from the expert and freeing up their cognitive resources for other tasks (e.g., identification of interesting, anomalous features in seismic images).

In our hypothetical example, a geoscientist analyzes a 3D seismic volume, interpreting in a series of 2D in-line images

faults and horizon off-sets. As subsequent in-lines (A-F) are interpreted, fault displacement patterns are co-visualized, so inconsistencies from normal fault displacement can be clearly seen. Fault 1 (Image B) conforms to a simple fault-displacement pattern (see Fault 1 displacement-distance plot). Fault 2 appears to conform to a similar pattern until in-line D when the inter-





preted displacement decreases; on interpretation of in-line E, the displacement on Fault 2 increases again, further highlighting the displacement anomaly on in-line D. Reduced displacement in itself does not highlight an issue, but consideration of the
displacement-distance plot for Fault 1 suggests that if the interpreted displacement for Fault 2 is correct then the two faults are behaving differently. In our imagined digital tool, this discrepancy in displacement between nearby faults would be flagged for further consideration by the user. You can see the hypothetical conclusion certainty plots for the interpreter for the two faults (Fault 1 = green line, Fault 2 = pale blue line) during the interpretation process. Note the decrease in certainty of the interpreter for Fault 2, as they interpret in-lines D and E, in comparison to the increasing certainty for Fault 1 as consecutive interpreted in-
lines conform to a simple normal fault displacement pattern. At in-line E the co-visualised displacement-distance plot nudges the interpreter to consider a new interpretation for Fault 2 at in-line D. Certainty in this new interpretation (displayed as dark blue dashed line on certainty plot), now increases as subsequent in-line interpretations conform to expected displacements.

Our imagined digital tool builds on current auto-correlation tools in seismic interpretation software by aiding users in ex-trapolating information from a 2D image to a 3D representation. Current auto-correlation tools allow unbiased quantitative
constraints to be placed on horizon interpretations, but look at each horizon pick in isolation and do not employ geologically reasonable tests across the broader dataset, e.g. they do not use information from horizon interpretations above and below to inform their choices, instead working only on the data provided. In this case study we show that by drawing on known fault displacement patterns, it should be possible to design tools that flag to users potential errors in fault displacement patterns along interpreted faults in 3D seismic data. We describe information for a single horizon displacement, but multiple horizons
could be plotted to highlight displacement changes with depth, syn-sedimentation etc. Our case study uses a simplified case, but fault displacement inconsistencies would likely be the result of more complex fault patterns and interactions. For example it is possible to imagine a scenario in which both Fault 1 and Fault 2 showed significant decreases in displacement at in-line D, which might result in a decrease in user certainty for both faults and re-interpretation as linked faults. As highlighted in a 3D seismic interpretation of faulted sedimentary rocks by Freeman et al. (2010), fault intersections are common and add challenges
to understanding fault growth and displacement partitioning between faults; in their example, a full reinterpretation of the 3D dataset was required after evaluation of fault displacements on the original interpretations. Therefore, a digital tool (similar to the one we describe) that highlights possible fault intersections and relays during interpretation, could cue researchers that more complex reasoning is needed, so that simple dominant models – which as identified by Alcalde et al. (2017b) often show anchor effects – are tempered by consideration of more complex fault patterns and displacements.

**5 Conclusions**

Uncertainty is an inherent challenge in geological reasoning. Over 50 years of cognitive research demonstrates that, when faced with uncertainty, people rely on intuitive heuristics that can arise rapidly and with minimal effort. While efficient and effective in many situations, heuristics do lead to predictable biases in decision making. We reviewed three biases that have been shown to influence geoscience experts: availability bias, framing bias, and anchoring bias. Bias can be overcome by engaging
deliberative cognitive processing resources, which work as an "editor" to modify or override faulty heuristic-responses. This





occurs either because the decision maker employs a strategy that activates deliberative processes, or because the environment is modified in such a way that the decision maker is "nudged" towards deliberative thinking. Because of the many barriers to success when debiasing is self-employed (e.g., not recognizing debiasing is needing, using the incorrect debiasing strategy, etc.), we strongly advocate adoption of the environment-modification (i.e., choice-architecture) approach. Further, we believe innovations in the use of information technology and AI in the geosciences can be leveraged to improve expert decision making, i.e., digital nudging. We discussed two case studies illustrating different applications of intelligent systems for the debiasing of geoscientific decision making. In each case study, debiased decision making was an emergent property of the coordinated and integrated processing of human-AI collaborative teams.

Our discussion of digital nudging in the geosciences highlighted the positive attributes of this debiasing approach, chiefly, that it provides relief from the cognitive constraints that lead to biased choice (and difficulty in effectively self-employing debiasing strategies), leaving the decision maker and their deliberative cognitive processing resources free to tackle other tasks. However, we would be remiss to not also caution against the potential pitfalls of the digital nudge. First, digital nudges can propagate existing biases (or introduce new ones) if they are poorly designed or trained using biased data. In a recent famous case, Amazon ceased testing an AI system used to evaluate job applicants after it was revealed to be gender biased, consistently penalizing applicant's who attended women's colleges or whose resume contained the word "women's" in some other capacity (e.g., women's book club). Similar gender and racial biases have been demonstrated in judicial (Skeem and Lowenkamp, 2016) and medical (Challen et al., 2019) decision AI. To avoid unintended bias, choice architects must have a well-defined goal for the nudge and a clear understanding of the decision process, including: identification of the critical actions involved in following through with the decision, identification of constraints to the achievement of each critical action, determination of the amount of attention devoted to each critical action and the decision process as a whole, determination of the amount of decision-relevant information gathered, and identification of the main heuristics and biases influencing the decision process (see Ly et al. (2013), Appendix 2 for a suggested list of questions choice architects should ask themselves when evaluating a decision process). Choice architects should also submit nudge designs to careful testing, paying special attention to factors unrelated to the nudge that might influence the results.

A second pitfall is that there is a risk of limited take up of digital nudges if they are perceived by the user as untrustworthy. To be effective, a nudge must address the particular bias an individual is experiencing – but people can differ in the biases they bring to a choice environment. Nudges that are viewed as inappropriate or misleading by the user may be ignored and mistrusted. Therefore, choice architects should be thoughtful in their selection of environments (i.e., employing nudges when there is consistency in the type of bias observed in a specific environment), and seek to design nudges that are effective against a range of biases. Special attention should also be paid to the relative "politeness" of nudges (Whitworth, 2005), i.e., does the nudge respect and not preempt user choice, does the nudge avoid pestering or interrupting the user unnecessarily. Nudges that make correct suggestions, but do so in an impolite or obtrusive manner will still be viewed as untrustworthy – we call this the "Mr. Clippy problem" in reference to the famously derided Microsoft Office assistant that took the form of an animated paper clip. Early users of Office will recall that Mr. Clippy popped-up uninvited, preemptively taking control of the cursor and demanding to help (his most famous line, "It looks like you're writing a letter..." appeared every time the user typed





"Dear. . . "). Worse yet, Mr. Clippy ignored continuous rejection: hide him and he would simply reappear, ignore him and he would repeat the unsolicited advice (again and again). To avoid the Mr. Clippy problem, choice architects should consider how

best to implement nudges within existing user workflows to minimize distraction and maintain user autonomy.

One way choice architects may increase understanding and trust of digital nudges is through being transparent in the "reasoning" behind why a nudge is prompted, where reasoning refers to some interpretable translation of the underlying AI algorithm and decision inputs. This type of "explainable AI" (Miller, 2019) is critical to our vision of collaborative and coordinated decision making in human-AI teams. Just as successful human teams are aware of the values, needs, intentions, actions, and

capabilities of all team members, so should human-AI teams be reciprocally aware – this occurs over time, through interaction, shared experience, and feedback, team members can jointly and iteratively refine their beliefs and expectations about each other's behavior. While there are challenges to achieving mutually aware human-AI teams (for review, see de Graaf and Malle (2017)), they are, in our opinion, far outweighed by the potential value in calibrating user trust in technology.

Finally, there are concerns about the ethics and morality of nudging (digital or otherwise). Some believe that nudging is

morally reprehensible because it patronizes the decision maker by assuming they are not capable of making the best choice for themselves (Gigerenzer, 2015). However, as discussed in this paper, there is strong scientific evidence that human decision makers are both (a) susceptible to cognitive bias across a range of choices and (b) struggle to successfully employ debiasing techniques to improve their judgment. We believe that if decision makers are aware of their vulnerabilities and shown the potential value of nudging (through education or experience with polite and explainable nudges), they may be less likely to

perceive nudges as condescending or infantilizing. Another oft-cited ethical concern is that nudged individuals will become used to being guided away from negative consequences resulting in a diminished ability to make good choices and assume responsibility for those choices (Bovens, 2009). Related to this, there is concern that the more use to nudging we become, the less we will be bothered by the introduction of more controlling or coercive techniques (Rizzo and Whitman, 2009). Yet, how decision makers respond to nudges in the long-term is an open empirical question. One possibility is that nudges have only

short-term effects, and as time goes on, the level of nudging required to retain this effect increases because decision makers habituate to the nudge. If this is the case, then decision makers would retain their original preference structures, meaning they would make different choices without the aid from the nudge, placing them at risk of taking less personal responsibility for their choices because they assume other members of society will nudge them away from anything that is bad. The alternative is that repeated nudges induce actual preference change in the long-term; this could occur because the decision maker recognizes the

hitherto unknown benefits of the nudged choice, because their sense of identity becomes linked to the nudged choice, or because the nudged choice becomes conditioned (in the Pavlovian style). Ultimately, different people will likely adapt preferences in response to different nudges in different ways, and future research should consider both the short-term and long-term effects.

Returning to the opening quote and question posed by G. K. Gilbert, "...whether it is possible by training to improve the guessing faculty, and if so, how it is to be done", the answer is unequivocally yes, and we believe that digital nudging offers the

best opportunity to overcome the cognitive constraints that result in biased decisions. As described at the outset of this paper, we hope our review of the cognitive literature on bias and debiasing will help readers to understand the constraints to human decision making and better-equip them with strategies for improving choice. We also hope this paper will stimulate future





research on the important topic of debiasing geologic decision making, particularly in the context of evolving advancements in information technology and AI.

*Author contributions.* All authors contributed to the preparation of this manuscript.

*Competing interests.* The authors declare that they have no conflict of interest.

*Acknowledgements.* C. G. Wilson and T. S. Shipley are currently supported by NSF Science of Learning Collaborative Network grant (1640800), NSF National Robotics Initiative grant (1734365), and NSF Future of Work at the Human Technology Frontier grant (1839705). C. E. Bond is currently supported by a Royal Society of Edinburgh research sabbatical grant.





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





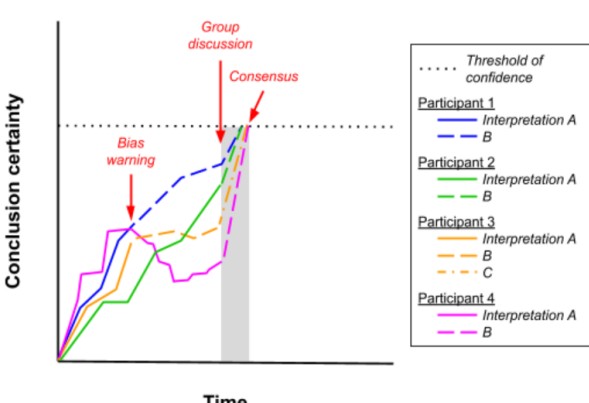

**Figure 3.** The influence of expert elicitation practices on conclusion certainty in a study by Polson and Curtis (2010). All participants experience shifting uncertainty in initial probabilities for the existence of a fault as elicitation progresses. Interpretation A, B, and C refer to each individual's successive interpretation. When exposed to information on how cognitive bias impacts on expert judgement, Participants 1 and 3 modify their interpretations (to 1B and 3B) and Participant 4 experiences a decline in conclusion certainty (4A). Participant 2, in contrast, becomes more certain in his or her initial interpretation following the bias warning, either because the interpretation was truly unbiased or because the warning was insufficient to recognize and resolve bias. When interpretations are shared and discussed amongst the group, Participants 2, 3, and 4 modify their interpretations to be in accordance with Participant 1 – certainty in this new interpretation (2B, 3C, 4B) increases such that a consensus assessment is reached. This herding bias is noted by the gray zone in the figure.

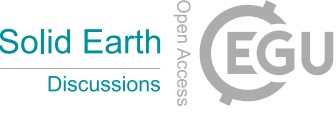



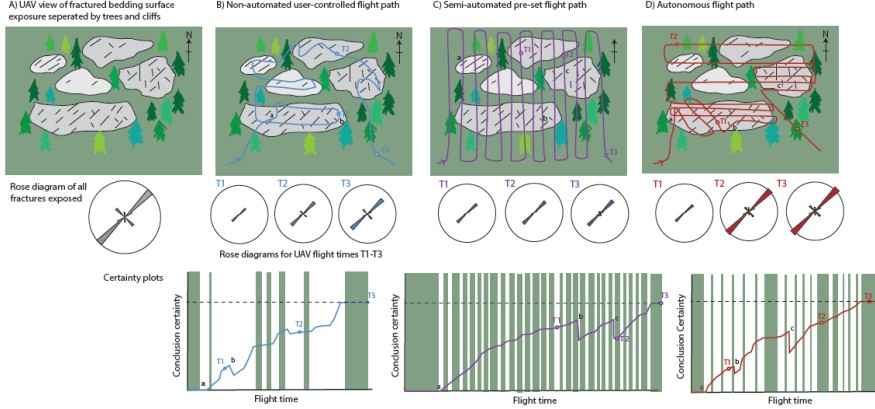

**Figure 4.** Hypothetical UAV scenario where the goal is to identify fractures on a bedding surface. Column A shows UAV view of fractured bedding surface (with exposure separated by trees and cliffs) and a rose diagram of all exposed fractures. Column B shows user-controlled flight path (in blue) over bedding surface, rose diagrams of cumulative fracture orientation data at flight times (T1, T2, T3), and an evolving conclusion certainty plot as the UAV collects data. On the conclusion certainty plot, green bars represent flight time over woodland rather than rock exposure, letters a, b, and c represent when the UAV collects fracture data in a new orientation, and the dotted horizontal line represents the threshold of conclusion confidence. Column C shows similar plots for a pre-set flight path (purple), and Column D shows plots for an autonomous flight path (red) where the UAV attends and reacts to measurement data in real-time.

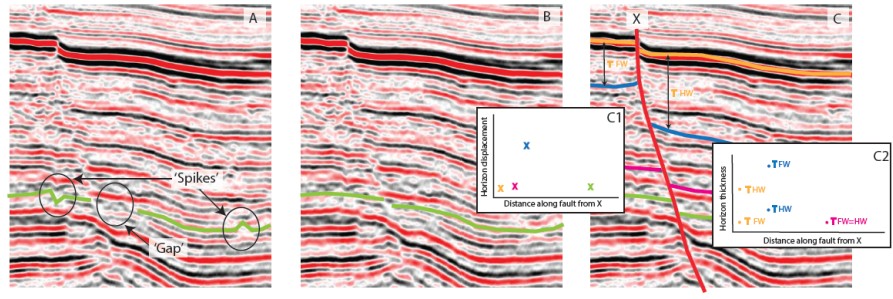

**Figure 5.** Hypothetical interpretations of horizons offset by a fault imaged in seismic data. Image A shows the interpretation of a (green) horizon, with "spikes" and "gaps" highlighted, as commonly seen in interpretations completed by auto-tracking. Image B shows the same horizon as in Image A, but cleaned and smoothed. On Image C, further horizon interpretations have been made, including identification of a fault. A point on the fault (X) is used for calculations in plots C1 and C2. C1 shows horizon displacement with distance along the fault from X. C2 shows horizon thickness along the fault, showing thickness in both the foot wall (FW) and hanging wall (HW) for each horizon – an example (orange horizon) of how the thickness measurements are made is shown on Image C. Seismic image courtesy of the Virtual Seismic Atlas (www.seismicatlas.org).





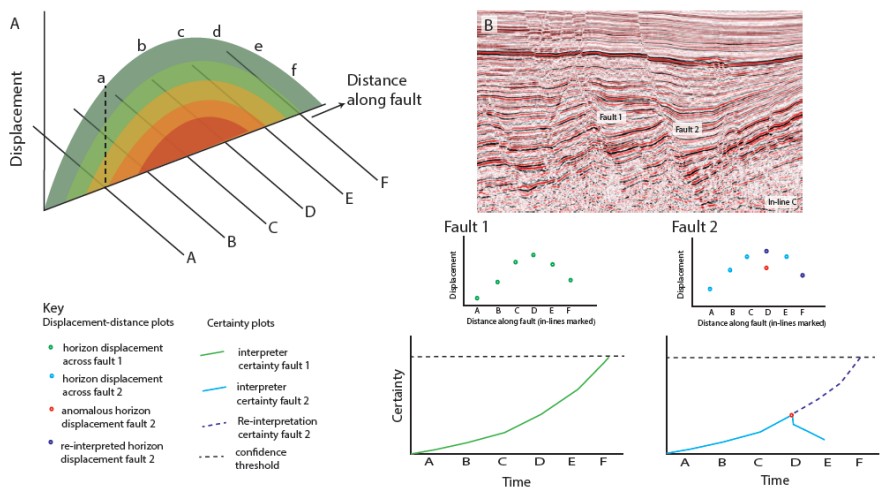

**Figure 6.** Fault displacement-distance patterns used by a hypothetical digital tool to aid in 3D interpretation of normal faults. Image A is a graphical representation of a horizon displacement with distance along a fault. In this simple pattern for an isolated normal fault, maximum displacement (red-green) is in the centre with minimal displacement (green) at the faults tips. The fault is intersected by a series of hypothetical seismic lines (A-F), that correspond to the points on the displacement-distance plots. Image B is a seismic image through faulted sedimentary rocks from the Inner Moray Firth, UK (note the complex fault pattern, including fault intersections). Two faults are highlighted on Image B, Fault 1 and Fault 2, their respective hypothetical displacement-distance plots are show for a single horizon. On the Fault 2 plot, the red point at in-line D highlights an anomaly to the simple displacement-distance plot characteristics seen for Fault 1, and the dark blue point at in-line D represents the user manually inserting a new interpretation. Certainty plots corresponding to Fault 1 and Fault 2 are also displayed. Seismic image courtesy of the Virtual Seismic Atlas (www.seismicatlas.org).