# Peer review of "How can geologic decision making under uncertainty be improved?"

_Solid Earth, 2019_

## Referee Comment (RC1) · Simon Oldfield (Referee) · 13 May 2019

OVERVIEW COMMENTARY

This manuscript represents a valuable contribution to the geological literature applying established psychological understanding of decision-making to the interpretation of uncertain geological information. I support its publication in Solid Earth as a means to highlight this field of research to geological interpreters in various sub-disciplines.

My overview comments below highlight some observations that are intended to provide an alternative perspective to the authors that may contribute to some improvements, though actions are not expected on all points. I would be pleased to discuss any of my comments further with the authors.

[Figure]

Throughout the manuscript the term decision-making is used in various ways. There may be benefit in clarifying the type of judgement (is it primarily subject to aleatory or epistemic uncertainty) and the sensitivity of the final outcome to that decision. This may allow clearer suggestion of which biases / debiasing strategies are most significant for that case. Furthermore this may add weight to your overall argument, identifying (in-line with past work) that higher impact (more summative) decisions, may be more vulnerable to significant impacts of human bias (Begg, Welsh, & Bratvold, 2014).

The case studies are well-written and demonstrative of discussed principles. The description of the geological issue could be shortened, while discussion of the psychological perspective and suitability of a particular approach to address the identified psychological issues raised should be enhanced.

Throughout the manuscript there are a number of comments regarding implementation of IT and AI solutions. These comments should be justified with expansion on the specific aspect of human bias or decision-making uncertainty that is being addressed, linking the comments back to the theme of the manuscript. Without this justification, the comments are slightly redundant and detract from the overall message.

DETAILED COMMENTARY

P. 1; ln. 1-16: Suggest the abstract should be broken into paragraphs, though perhaps this is a formatting error within the manuscript submission process (?).

P. 2; ln.11: Repetition of preservation / exposure issues from ln. 8-9.

P.3 ln. 7: Are there any further comments to be made on biases that have not already been considered in the geological literature? Is there value in future work on these themes?

P. 3; ln. 18: May benefit from clarification of how 'maximising the utility of a decision' relates to a problem where your aim is to best characterise a system, rather than maximise or minimise an individual parameter such as volume or cost.

P. 3; ln. 14: It may be useful to define decision-making and discriminate between decisions at different levels, i.e. calculation of individual parameters versus overall interpretations, throughout the manuscript. In the latter elements this would enable more seamless reference back to earlier points of discussion.

P. 3; ln. 15: Normative decision models. Though the current examples are clear for a non-specialist reader, may benefit from direct relationship of these principles to a geoscience problem featuring a sparse and irregularly sampled time series (less common with the normal economic and financial examples).

P. 4; ln. 3: Completely agree with the direction of your argument here, however it remains ambiguous, geoscientists can make optimal choices, however it may be worth noting that an optimal choice may be consideration of multi-scenario interpretations (common in the hydrocarbons industry).

P. 5; ln. 18: Perhaps this line could be framed as a description of what is to follow, rather than a continued description of the intuitive and deliberative causes.

P. 7; ln. 4: Arguably, students may have an advantage in some of these settings as they may have more recently experienced a broader range of geological settings and applications.

P. 8; ln. 10: Great to see this highlighted, however are the experts addressing epistemic or aleatory uncertainty in this exercise? The prediction of the range of variability could indicate that this would be an aleatory uncertainty, unless guided by an underlying epistemic uncertainty in the interpreted mechanism of corrosion.

P. 9; ln. 28: Multi-scenario modelling is common in some sub-disciplines, making the general application of this statement to the whole discipline of geology inaccurate.

P. 11; ln. 34: Which models are referred to here (i.e. mental, geological, risk-models)?

P11; ln. 13-16: Personal opinion: This highlights the importance of sensitivity analysis, adding a sensitivity analysis to the end of any workflow providing immediate feedback

to the practitioner (/learner).

P.12; ln. 20: Much of the uncertainty training available to industry focusses on understanding probability and raises awareness of bias in this context. In reality this is normally coupled with training in a structured workflow, potentially negating the implications of Fischoff's description despite its broader relevance.

P. 12; ln. 22: Depending on the location of this box, a definition of choice architecture and nudging may be required.

P. 12; ln. 30: The description of choice architecture should be clarified with a more concise description.

P. 13; ln. 7: Suggested that this paragraph be broken up.

P. 13; ln. 20: A comment or citation with regards to standardisation of workflows in petroleum engineering would be well placed here.

P. 14; ln. 35: Check for the plurality of data.

P. 14; ln. 37: Could a steeper line also indicate a simpler problem?

P. 16; ln. 9-11: On expert elicitation, it may be useful provide examples of how statistical approaches could be used to guide appropriate filtering and averaging of expert opinions (e.g. Aspinall, 2010).

P. 17; ln. 6: Considering this section; the principle advantage to this approach seems to be the reduction in cognitive load on the geoscientist during acquisition freeing up time for field location interpretation and potentially making a more uniform/complete dataset. Reading this section, I felt the psychological advantage was slightly disjointed compared to the psychological challenges (including bias) discussed earlier in the manuscript. A sentence framing the issue in line with the intuitive / deliberative thinking (P. 4; ln. 15) may provide a route to achieving this?

P. 19; ln. 16: Could the sub-titles of the case studies be altered slightly to indicate

their psychological significance. e.g. Case study 2: Automation as a tool to allow more deliberative thinking. Case study 2: A nudge to verify fault interpretation.

P. 19; ln. 21: The analysis/interpretation of seismic volumes is inherently 3D, even if visualisation is in 2D. Methods such as 3D geobody extraction, geometric attribute analysis or horizon extractions are commonly employed and 3D in both analysis and visualisation. Description of the geological technique could potentially be summarised, allowing further discussion of the psychological challenges to be addressed.

P. 20; ln 19: Nice example of using nudges to influence workflow decision. Again, description of the geological technique could potentially be summarised, allowing further discussion of the psychological challenges to be addressed.

REFERENCES

Aspinall, W. (2010). A route to more tractable expert advice. Nature, 463(January), 294–295.

Begg, S. H., Welsh, M. B., & Bratvold, R. B. (2014). Uncertainty vs. variability: What's the difference and why is it important? In SPE Hydrocarbon Economics and Evaluation Symposium. Houston, Texas: Society of Petroleum Engineers.

Bentley, M., & Smith, S. (2008). Scenario-based reservoir modelling: the need for more determinism and less anchoring. Geological Society, London, Special Publications, 309(1), 145–159.

---

## Referee Comment (RC2) · Florian Wellmann (Referee) · 5 Jul 2019

In the manuscript with the title "How can geologic decision making under uncertainty be improved?", the authors present an overview of the state of the art of decision making and the three main forms of bias, which are typical in geological investigations. In addition, the authors describe approaches to optimise decision making based on debiasing methods using "digital nudging" approaches. The manuscript is an important contribution to the topic of uncertainty estimation in geological studies and well suited for publication in Solid Earth.

The first part of the manuscript addresses availability, framing, and anchoring bias as common forms of bias in our field. This information is mostly a review of existing work,

but an important compilation on the topic, including recent references and placing them in the geological context.

Very interesting is also the section on debiasing strategies and on the question of methods to teach decision making (in a geological context).

The second part of the manuscript is then focussed on two case studies on the combination of "AI" with human interpretations in order to improve decision making. In this part, I have some problems in following the argumentation of the authors. I understand that any help with reducing cognitive overloading ("busy editor") can potentially help. But especially in the first case, I do not quite see how an automated sampling strategy can help here. For sure, an optimised sampling is interesting in itself - but how does this address the three forms of bias presented above, as opposed to a pure random sampling or the commonly used regular flight paths (option "C" in Fig. 4)? The only added benefit I see (maybe because this is a simplified example) is the reduced time of sampling. Even more: couldn't one also argue that any form of "AI" is prone to introducing additional bias, as it is based on an underlying algorithm that may also be biased? Also, you argue in line 13 (pg. 19) that the expert (user) should retain the ability to interact and adjust the flight path - but wouldn't this then again be prone to the biases described before?

The aspect of fault interpretations in seismic data, explained in case study 2, is more obvious to me - although here the question could also be how much bias is in the initial choice of a fault displacement model (which can be based on physical principles, but the potential interactions can also quickly become very complex when considering fault networks, relay structures, etc.). But here, the point of flagging potential areas of problems is an interesting aspect of "digital nudging" (if I understand it correctly), and similar to the example from Polson and Curtis (2010) and the "bias warning" point in the expert decision-making process.

In summary, even with my (minor) comments on the case studies above, where some

clarification could be beneficial, the manuscript has many very interesting and thought-provoking sections that add important aspects to the discussion on uncertainties in geosciences and I am looking forward to seeing it published.

---

## Referee Comment (RC3) · Anonymous Referee #3 · 11 Jul 2019

This manuscript discusses a very interesting and important topic, namely how to improve decision making processes in the geosciences. It does a very nice job of reviewing the most common bias types known to affect geologists, giving excellent examples that make it easy to learn about those bias types. I found Section 1-3 a pleasure to read, learned a lot, and thought that this would be excellent reading for scientists in our field.

However, Section 4 did not convince me. I still followed the arguments in Sections 4.1 and 4.2, describing why it is more successful to work on changing the environment, rather than teaching the decision maker new skills. But I found it hard to truly understand how the two case studies in Section 4.2 truly connect with the biases outlined in the earlier sessions. In particular, Case 2, which focuses on sub-surface geology had

so many details related to sub-surface geology that I found it generally hard to follow. Both cases seemed to be examples that show that one should provide the user with as much useful information and software support as possible. Obviously, it's always useful to provide as much reliable, comprehensive information as possible to a decision maker, and such information will improve the decision process, and the software should be as transparent as possible. What am I missing here? Maybe the new point is that the biases are to be detected and software is to be designed specifically to overcome these specific biases. Is that the key point? But how do you identify all these holes and biases? How do you design software to fill these holes? How do you make sure the software solution is reliable in all cases and does not "nudge" the expert into the wrong correction?

In terms of style, the first 3 sections are a pleasure to read, but include some repetitions, e.g., many statements are made in Section 1 and then repeated in more detail in later sections. So I suggest to look for redundant statements and shorten those sections a bit. For Section 4 my main suggestions would be to 1) work hard on clarity in the case studies of how exactly they connect to the biases discussed earlier; 2) spell out the way forward, i.e. how could the lessons learned here be generalized to other applications.

In fact, I might even suggest to drastically shorten + de-emphasize the case studies to be only 1-2 paragraphs each, and maybe moving the rest to an appendix, then to focus on the main message of the paper in terms of fighting general biases and how to do that instead. I don't know, however, whether the remaining material warrants publication in this venue.

Minor comments: P. 5, Line 16-17: Mentions the three types of biases for the 3rd time. Too much repetition. P. 5, Line 30. There's something missing here. "Over ??? have found ... " P. 11. There is a Section 4.1.1, but no 4.1.2. P. 14, Line 24. "of of"

So, in summary, I really like the general topic and coverage of the paper, and think

topics like this are truly important to bring to the forefront, discuss and find solutions for. Section 1-3 are excellent for this purpose. However, I find Section 4.2 in its current form rather confusing, so IMHO the way forward is not clear.

---

## Author Response (AR1)

**Simon Oldfield**

Throughout the manuscript the term decision-making is used in various ways. There may be benefit in clarifying the type of judgement (is it primarily subject to aleatory or epistemic uncertainty) and the sensitivity of the final outcome to that decision. This may allow clearer suggestion of which biases / debiasing strategies are most significant for that case. Furthermore this may add weight to your overall argument, identifying (in-line with past work) that higher impact (more summative) decisions, may be more vulnerable to significant impacts of human bias (Begg, Welsh, & Bratvold, 2014).

- In this paper we use the term decision making broadly, to refer to any selection of an action/belief among alternative possibilities. Also, as discussed in the first paragraph, we are primarily focused on epistemic uncertainty. However, aleatory variability may still play a role in decision making since *judgments* (i.e., assessments of probability of an event) and *predictions* (i.e., judgments about uncertain states) are inputs for decision making. Ultimately, we feel that providing this level of nuance (in the definition of decision making) to the reader is unnecessary and may even confuse our message.
- No changes to manuscript

The case studies are well-written and demonstrative of discussed principles. The description of the geological issue could be shortened, while discussion of the psycho- logical perspective and suitability of a particular approach to address the identified psychological issues raised should be enhanced.

- To address reviewer comments about the case studies, we have shortened the geologic descriptions (removing sub-discipline jargon) and improved the connection between the case study section and the preceding bias review section. Details of these changes are reviewed in comments below.

Throughout the manuscript there are a number of comments regarding implementation of IT and AI solutions. These comments should be justified with expansion on the specific aspect of human bias or decision-making uncertainty that is being addressed, linking the comments back to the theme of the manuscript. Without this justification, the comments are slightly redundant and detract from the overall message.

- We mention IT/AI solutions for debiasing early in the paper to foreshadow our two case studies, which demonstrate the potential value of IT/AI for decision making. However, we agree that some mentions of IT/AI are repetitive and (without the justification that comes later in section 4.2) do not benefit our overall message.
- We removed one repetitive reference to IT/AI in the Introduction (P2, L35), so IT/AI solutions to debiasing are only mentioned once in the final paragraph (P3, L11). We removed another reference to IT/AI at the start of the debiasing section (P9, L7). IT/AI solutions are not mentioned again until just prior to the introduction of the case studies, where we describe why IT/AI solutions have great debiasing potential (P16, L20).

P. 1; In. 1-16: Suggest the abstract should be broken into paragraphs, though perhaps this is a formatting error within the manuscript submission process (?).

- This is not a formatting error. We prefer the presentation of the abstract as a single paragraph because the paper is primarily offering an approach to thinking about the human component of decision making, and not a more traditional research report with distinct methods and findings.
- No changes to manuscript

P. 2; ln.11: Repetition of preservation / exposure issues from ln. 8-9.
- Agreed that this is repetitive. The sentence has been modified to remove the repetition.
- "e.g. where isolation of processes can be difficult because multiple processes have cumulatively transformed the rocks, and where direct observation (much less experimental control) is impossible due to the large time spans of geologic processes, which leaves evidence lost or buried beneath the Earth's surface"

P.3 ln. 7: Are there any further comments to be made on biases that have not already been considered in the geological literature? Is there value in future work on these themes?
- We believe there is value in establishing what biases (in addition to framing, anchoring, and availability) may be impacting geologic decision making, although it is not the goal of the present manuscript. We make this point on P2, L24: "Characterizing the impact of decision biases such as the availability bias is important and more work is needed to determine the range of biases influencing geoscientists and their prevalence in geologic decision making." We also refer to reviews of biases, so readers know where to go to get more information on P5, L19: "These three biases by no means exhaust the full range of biases that could be influencing geologic decision making under uncertainty, but they are, at present, the best-documented in the geosciences literature. For a more complete list of biases and their potential influence on geologic decision making see Baddeley, Curtis, & Wood (2004), Bond (2015), and Rowbotham et al. (2010)."
- No changes to manuscript

P. 3; ln. 18: May benefit from clarification of how 'maximising the utility of a decision' relates to a problem where your aim is to best characterise a system, rather than maximise or minimise an individual parameter such as volume or cost.
- To address this comment (and the following one, and P4 L3 below), we included text clarifying that "optimality" can be defined in scientific decisions occurring at multiple levels, and normative models are a benchmark for assessing optimality across levels.
- P3, L16: "What does it mean to choose optimally during scientific decision making? The scientific decision process is complex and dynamic, and "optimality" may be defined at various levels, from selection of measurement tool and sampling site, to calculation of individual parameters, to interpretation (single or multi-scenario). The position we take in this article is that normative decision models offer a reasonable benchmark for assessing optimal choice-in geoscience decisions at all levels."

P. 3; ln. 14: It may be useful to define decision-making and discriminate between decisions at different levels, i.e. calculation of individual parameters versus overall interpretations,

throughout the manuscript. In the latter elements this would enable more seamless reference back to earlier points of discussion.

- To address this comment, we included text clarifying that "optimality" can be defined in scientific decisions occurring at multiple levels, and normative models are a benchmark for assessing optimality across levels.
- P3, L16: "What does it mean to choose optimally during scientific decision making? The scientific decision process is complex and dynamic, and "optimality" may be defined at various levels, from selection of measurement tool and sampling site, to calculation of individual parameters, to interpretation (single or multi-scenario). The position we take in this article is that normative decision models offer a reasonable benchmark for assessing optimal choice-in geoscience decisions at all levels."

P. 3; ln. 15: Normative decision models. Though the current examples are clear for a non-specialist reader, may benefit from direct relationship of these principles to a geoscience problem featuring a sparse and irregularly sampled time series (less common with the normal economic and financial examples).

- The purpose of section 2 is to give some history on how optimality has been traditionally considered (in economic/financial domains). We relate this traditional view of optimality to the domain of geoscience in sections 3.1 through 3.3 when we discuss specific biases (availability, framing, anchoring).
- No changes to manuscript

P. 4; ln. 3: Completely agree with the direction of your argument here, however it remains ambiguous, geoscientists can make optimal choices, however it may be worth noting that an optimal choice may be consideration of multi-scenario interpretations (common in the hydrocarbons industry).

- To address this comment, we included text clarifying that "optimality" can be defined in scientific decisions occurring at multiple levels, and normative models are a benchmark for assessing optimality across levels.
- P3, L16: "What does it mean to choose optimally during scientific decision making? The scientific decision process is complex and dynamic, and "optimality" may be defined at various levels, from selection of measurement tool and sampling site, to calculation of individual parameters, to interpretation (single or multi-scenario). The position we take in this article is that normative decision models offer a reasonable benchmark for assessing optimal choice-in geoscience decisions at all levels."

P. 5; ln. 18: Perhaps this line could be framed as a description of what is to follow, rather than a continued description of the intuitive and deliberative causes.

- The aim of this section is to describe the mechanism of decision biases, which involves the interaction of intuitive and deliberative processes. We believe it is important to highlight here how the three biases to be discussed arise from these processes. We cleaned up the text to make it more readable.
- "All three are driven by faulty heuristic responses, which should be overridden by deliberative processes but are not. A form of anchoring bias can also be driven by flawed deliberative processing, which is discussed."

P. 7; ln. 4: Arguably, students may have an advantage in some of these settings as they may have more recently experienced a broader range of geological settings and applications.

- This would certainly be likely to aid them with availability bias (which arises because of reliance on salient cases from memory), but would have less of an impact on framing bias (which arises from emotional reactions to positive/negatives). Although it is not a question we address in this paper, it is interesting to consider how level of domain expertise influences decision making, i.e., in what situations is expertise a help versus a hindrance?
- No changes to manuscript

P. 8; ln. 10: Great to see this highlighted, however are the experts addressing epistemic or aleatory uncertainty in this exercise? The prediction of the range of variability could indicate that this would be an aleatory uncertainty, unless guided by an underlying epistemic uncertainty in the interpreted mechanism of corrosion.

- Phillips (1999) assesses epistemic uncertainty, probability distributions were obtained from experts who considered an uncertain quantity relevant to risk analysis of a proposed nuclear waste facility.
- No changes to manuscript

P. 9; ln. 28: Multi-scenario modelling is common in some sub-disciplines, making the general application of this statement to the whole discipline of geology inaccurate.

- In our understanding, multi-scenario modeling is a form of computational modeling. We view computational modeling as distinct from Chamberlin's method of multiple working hypotheses, which describes the process of contrasting mental models. We discuss the value of computational models in the paragraphs following.
- No changes to manuscript

P. 11; ln. 34: Which models are referred to here (i.e. mental, geological, risk-models)?

- Here we refer to computational models. Clarification added.
- "there is concern amongst geoscience scholars that decision makers are not using computational models as often as expected, or correctly"

P11; ln. 13-16: Personal opinion: This highlights the importance of sensitivity analysis, adding a sensitivity analysis to the end of any workflow providing immediate feedback to the practitioner (/learner).

- We agree!
- No changes to manuscript

P.12; ln. 20: Much of the uncertainty training available to industry focuses on un- derstanding probability and raises awareness of bias in this context. In reality this is normally coupled with training in a structured workflow, potentially negating the implications of Fischoff's description despite its broader relevance.

- It is encouraging to know that workflows are incorporated in training. We included a sentence describing this common practice.
- "Trainings on using structured workflows, as are common in geoscience industry, is one existing method of incorporating choice architecture techniques in education."

P. 12; ln. 22: Depending on the location of this box, a definition of choice architecture and nudging may be required.
- Noted. For the proposed layout we anticipate that that will not be necessary.
- No changes to manuscript

P. 12; ln. 30: The description of choice architecture should be clarified with a more concise description.
- Agreed this needs to be more concise. We edited the paragraph intro to improve clarity.
- "Debiasing techniques that modify the environment alter the settings where decisions occur. The environment can be modified to make it a better fit for the strategy people naturally apply (e.g., status-quo bias pushes people to stick with a default response option over selecting a new option, so making the default a desirable outcome will maximize decision making). The environment can also be modified to "nudge" people towards the optimal choice strategy (e.g., prompts to induce reflection and deliberation)  This environment modification approach to debiasing is sometimes referred to as *choice architecture*, making the individual or entity responsible for organizing the environment in which people make decisions the *choice architect* (Thaler & Sunstein, 2009)."

P. 13; ln. 7: Suggested that this paragraph be broken up.
- Agreed this paragraph needed to be broken up.
- A new paragraph was started with "The advantage of debiasing techniques that modify the environment, over those that modify the decision maker, is that it is the choice architect and not the decision maker who is accountable for debiasing (unless, of course, the architect and the decision maker are the same person).."

P. 13; ln. 20: A comment or citation with regards to standardisation of workflows in petroleum engineering would be well placed here.
- Our understanding is that the standardization of workflows in petroleum geology and engineering was designed to ease comparison between researcher reports within the same company, and not to aid researcher decision making (our focus here). Therefore, we don't think a reference here is appropriate.
- No changes to manuscript

P. 14; ln. 35: Check for the plurality of data.
- Noted.
- Changed from "data is" to "data are"

P. 14; ln. 37: Could a steeper line also indicate a simpler problem?

- Absolutely – and our definition of an "ideal" case is really just the case of a simple scientific problem. In other words, simpler scientific problems should be solved more efficiently than complex problems.
- No changes to manuscript

P. 16; ln. 9-11: On expert elicitation, it may be useful provide examples of how statistical approaches could be used to guide appropriate filtering and averaging of expert opinions (e.g. Aspinall, 2010).
- We reference the importance of appropriate filtering/averaging in the next paragraph when we state, " For example, research on expert elicitation practices in the geosciences has shown that erroneous predictions about geologic events are made when using subjective methods for selecting experts (Shanteau, Weiss, Thomas, & Pounds, 2002), and when judgments are not aggregated appropriately (Lorenz, Rauhut, Schweitzer, & Helbing, 2011; Randle et al., 2019). " We feel a more thorough discussion of aggregating methods is beyond the scope of this paper.
- No changes to manuscript

P. 17; ln. 6: Considering this section; the principle advantage to this approach seems to be the reduction in cognitive load on the geoscientist during acquisition freeing up time for field location interpretation and potentially making a more uniform/complete dataset. Reading this section, I felt the psychological advantage was slightly dis- jointed compared to the psychological challenges (including bias) discussed earlier in the manuscript. A sentence framing the issue in line with the intuitive / deliberative thinking (P. 4; ln. 15) may provide a route to achieving this?
- We have improved the connection between the case study section and the preceding bias review section by explicitly detailing how automated flights can address susceptibility to anchoring bias in field decisions about where to fly. The title of case study 1 was changed to: "*Optimizing field data collection with UAVs to minimize anchoring bias*". We have also removed many extra details about UAVs not directly pertinent to the case study. Most of the changes were to the first three paragraphs
- "In this case study, we describe how automated UAV navigation could be used to nudge geoscientists to be more efficient when making decisions regarding reconnaissance and mapping and mitigate against anchoring bias. The advent of better mobile robot platforms has allowed for the deployment of robots by ground, sea, and air to collect field data at a high spatial and temporal resolution. Here, we focus on the use of aerial robots (semi-autonomous or autonomous UAVs) for data collection, but the conclusions we draw are likely applicable to other mobile robot platforms (i.e., underwater autonomous vehicles, ground robots).
- Currently, the majority of geoscience research with UAVs is non-autonomous, i.e., user-controlled. Efforts have been made to automate interpretation of geological data from UAV imagery or 3D reconstruction with some success (Thiele et al., 2017; Vasuki, Holden, Kovesi, & Micklethwaite, 2014; Vasuki, Holden, Kovesi, & Micklethwaite, 2017), and the application of image analysis and machine learning techniques continue to

be developed (Zhang, Wang, Li, & Han, 2018). In reconnaissance and geologic mapping, the decision of where to go and how to fly there is made by the expert – either the expert fly's the UAV and makes navigation decisions in-situ or they pre-set a flight path for the UAV to follow semi-autonomously (cf. Koparan et al., 2018; Ore, Elbaum, Burgin, & Detweiler, 2015). However, a UAV that is capable of attending to measurements in real time and reacting to local features of measurement data could navigate autonomously to collect observations where they are most needed. Such autonomous workflows should increase the efficiency of data collection, and could be designed to mitigate against potential biases. Here, we consider how an automated UAV navigation nudge could reduce the tendency to anchor field exploration based on existing models and hypotheses

- In our hypothetical example, a UAV surveys a large bedding surface with the aim of identifying fracture orientations. The bedding surface exposure is large, but split into difficult to access exposure, e.g., due to cliff-sections or vegetation (see Column A, Figure 4). A birds-eye view afforded by the UAV improves the ability to observe fractures, which would otherwise require time-costly on-foot reconnaissance to different outcrops of the bedding surface. Note that in our hypothetical example we assume that fracture information is obtained only when the flight path crosses fractures (e.g., Column B, blue flight path), thereby representing a high level reconnaissance rather than a flight path in which overlapping imagery is collected. When the UAV flight path is user-controlled, the decision of where and how to fly is unlikely to be optimal: users could be distracted by irrelevant information in UAV view, and are likely biased towards exploring certain features and ignoring others (see Andrews et al., 2019). For example fractures may only be sampled where fracture data is dense, or in an orientation that maximizes sample size but not the range in orientation (see Watkins, Bond, Healy, & Butler, 2015), or when it fits with a hypothesis (e.g. tensional fractures parallel to the axial trace of a fold). These strategies are all informed by expectations, leaving the geoscientist vulnerable to anchoring her sampling behavior to align with initial interpretations and hypotheses. This anchoring bias is visualized in Column B (blue flight path), where the user detects two unique fracture orientations (a, b) on the first exposure visited, but then spends needless time (T1 to T2) at exposure that offers no new information, before finally visiting exposure that features the previously identified orientations (a, b) and a novel N-S fracture orientation. This novel orientation is not detected in the user's flight path – the accompanying certainty plot in Column B shows that time spent at uninformative exposure (T1 to T2) results in increased certainty that all orientations have been sampled, when in fact they have not (i.e., the threshold of confidence is reached before sampling the N-S orientation). This is reflected in the rose diagrams in Column B, which show the orientation of fractures and the relative number of fractures sampled in each orientation; even at time T3 the three fracture sets (as shown in the rose diagram in Column A) are not represented."

P. 19; ln. 16: Could the sub-titles of the case studies be altered slightly to indicate their

psychological significance. e.g. Case study 2: Automation as a tool to allow more deliberative thinking. Case study 2: A nudge to verify fault interpretation.

- We agree that changing the titles to better reflect their psychological significance was needed.
- Case study 1 was changed to: "*Optimizing field data collection with UAVs to minimize anchoring bias.*", and Case study 2 was changed to: "*Fault interpretations in 3D seismic image data to minimize availability bias.*"

P. 19; ln. 21: The analysis/interpretation of seismic volumes is inherently 3D, even if visualisation is in 2D. Methods such as 3D geobody extraction, geometric attribute analysis or horizon extractions are commonly employed and 3D in both analysis and visualisation. Description of the geological technique could potentially be summarised, allowing further discussion of the psychological challenges to be addressed.

- We have improved the connection between the case study section and the preceding bias review section by explicitly detailing how seismic interpretation aids (built into software) can address susceptibility to availability bias during interpretation. The title of case study 2 was changed to: "*Fault interpretations in 3D seismic image data to minimize availability bias*". We also removed details about the technique of automated horizon tracking (including the accompanying Figure 5) which are not directly pertinent to the case study. Most of the changes were to the first three paragraphs
- "In this case study, we consider how software interpretations of seismic image data, and the information derived from them, could be used to nudge geoscientists to consider alternative models and minimize availability bias. Understanding of the geometries of sub-surface geology is dominated by interpretations of seismic image data, and these interpretations serve a critical role in important tasks like resource exploration and geohazard assessment. 3D seismic image volumes are analyzed as sequences of 2D slices. Manual interpretation involves visually analyzing a 2D image, identifying important patterns (e.g., faulted horizons, salt domes, gas chimneys) and labeling those patterns with distinct marks or colors; then, keeping this information in mind while generating expectations about the contents of the next 2D image. Given the magnitude and complexity of this task, there has been a strong and continued interest in developing semi-autonomous and autonomous digital tools to make seismic interpretation more efficient and accurate (e.g., Araya-Polo et al., 2017; Di, 2018; Farrokhnia, Kahoo, & Soleimani, 2018).
- Here, we consider how 3D information could be used with digital nudge technology to inform fault interpretations in a 3D seismic image volume. Simple normal fault patterns show a bull's-eye pattern of greatest displacement in the center of an isolated fault, decreasing towards the fault-tip (see Image A, Figure 6). Consider interpreting 2D seismic image lines across the fault starting at in-line A (Image A) and working towards in-line F: with each subsequent line the displacement of horizons across the fault should increase and then decrease, although this pattern will not be known until the interpretation is completed. Holding this information on displacements for individual

faults between in-line interpretations in complicated seismic image data (e.g. with multiple faults per seismic section, Image B, Figure 6) is incredibly challenging even for the well-practiced expert. We imagine a digital nudge that alerts users to discrepancies in fault displacement patterns, and prompts consideration of alternative fault patterns, thereby relieving some of the cognitive burden of 3D interpretation from the expert and guarding them against availability bias by encouraging consideration of models beyond what is most readily accessible to the mind.

- In our hypothetical example, a geoscientist analyzes a 3D seismic volume, interpreting in a series of 2D in-line images faults and horizon off-sets. As subsequent in-lines (A-F) are interpreted, fault displacement patterns are co-visualized, so inconsistencies from normal fault displacement can be clearly seen. Fault 1 (Image B) conforms to a simple fault-displacement pattern (see Fault 1 displacement-distance plot). Fault 2 appears to conform to a similar pattern until in-line D when the interpreted displacement decreases; on interpretation of in-line E, the displacement on Fault 2 increases again, further highlighting the displacement anomaly on in-line D. Reduced displacement in itself does not highlight an issue, but consideration of the displacement-distance plot for Fault 1 suggests that if the interpreted displacement for Fault 2 is correct then the two faults are behaving differently. In our imagined digital tool, this discrepancy in displacement between nearby faults would be flagged for further consideration by the user, and potential alternative models could be highlighted. You can see the hypothetical conclusion certainty plots for the interpreter for the two faults (Fault 1 = green line, Fault 2 = pale blue line) during the interpretation process. Note the decrease in certainty of the interpreter for Fault 2, as they interpret in-lines D and E, in comparison to the increasing certainty for Fault 1 as consecutive interpreted in-lines conform to a simple normal fault displacement pattern. At in-line E the co-visualized displacement-distance plot nudges the interpreter to consider a new interpretation for Fault 2 at in-line D. Certainty in this new interpretation (displayed as dark blue dashed line on certainty plot), now increases as subsequent in-line interpretations conform to expected displacements."

P. 20; ln 19: Nice example of using nudges to influence workflow decision. Again, de- scription of the geological technique could potentially be summarised, allowing further discussion of the psychological challenges to be addressed.

- See previous comment about case study 2.

**Florian Wellmann**

The second part of the manuscript is then focused on two case studies on the combination of "AI" with human interpretations in order to improve decision making. In this part, I have some problems in following the argumentation of the authors. I understand that any help with reducing cognitive overloading ("busy editor") can potentially help. But especially in the first case, I do not quite see how an automated sampling strategy can help here. For sure, an optimised sampling

is interesting in itself - but how does this address the three forms of bias presented above, as opposed to a pure random sampling or the commonly used regular flight paths (option "C" in Fig. 4)? The only added benefit I see (maybe because this is a simplified example) is the reduced time of sampling. Even more: couldn't one also argue that any form of "AI" is prone to introducing additional bias, as it is based on an underlying algorithm that may also be biased? Also, you argue in line 13 (pg. 19) that the expert (user) should retain the ability to interact and adjust the flight path - but wouldn't this then again be prone to the biases described before?

- We have improved the connection between the case study section and the preceding bias review section by explicitly detailing how automated flights can address susceptibility to anchoring bias in field decisions about where to fly. The title of case study 1 was changed to: "Optimizing field data collection with UAVs to minimize anchoring bias". We have also removed many extra details about UAVs not directly pertinent to the case study. Most of the changes were to the first three paragraphs
- Regarding AI introducing additional bias – Yes, this is possible, dangers and necessary precautions (i.e., explainability) are discussed in the conclusion.
- Regarding the user retaining autonomy to make biased decisions – this is a classic principle of the choice architecture approach, i.e., freedom of choice must never be encroached upon. As we state in section 4.2, "It is the role of the choice architect…to influence people's decision making such that their well-being (and the well-being of others) is maximized, without restricting the freedom to choose. Importantly, there is no such thing as neutral choice architecture; the way the environment is setup will guide decision making, regardless of whether the setup was intentional on the part of the architect, e.g., descriptions of risk will be framed in terms of gains or losses, a wise architect chooses the framing that will maximize well-being."

The aspect of fault interpretations in seismic data, explained in case study 2, is more obvious to me - although here the question could also be how much bias is in the initial choice of a fault displacement model (which can be based on physical principles, but the potential interactions can also quickly become very complex when considering fault networks, relay structures, etc.). But here, the point of flagging potential areas of problems is an interesting aspect of "digital nudging" (if I understand it correctly), and similar to the example from Polson and Curtis (2010) and the "bias warning" point in the expert decision-making process.

- We have improved the connection between the case study section and the preceding bias review section by explicitly detailing how seismic interpretation aids (built into software) can address susceptibility to availability bias during interpretation. The title of case study 2 was changed to: "Fault interpretations in 3D seismic image data to minimize availability bias". We also removed details about the technique of automated horizon tracking (including the accompanying Figure 5) which are not directly pertinent to the case study. Most of the changes were to the first three paragraphs

**Anonymous Referee**

Section 4 did not convince me. I still followed the arguments in Sections 4.1 and 4.2, describing

why it is more successful to work on changing the environment, rather than teaching the decision maker new skills. But I found it hard to truly understand how the two case studies in Section 4.2 truly connect with the biases outlined in the earlier sessions. In particular, Case 2, which focuses on sub-surface geology had so many details related to sub-surface geology that I found it generally hard to follow. Both cases seemed to be examples that show that one should provide the user with as much useful information and software support as possible. Obviously, it's always useful to provide as much reliable, comprehensive information as possible to a decision maker, and such information will improve the decision process, and the software should be as transparent as possible. What am I missing here? Maybe the new point is that the biases are to be detected and software is to be designed specifically to overcome these specific biases. Is that the key point? But how do you identify all these holes and biases? How do you design software to fill these holes? How do you make sure the software solution is reliable in all cases and does not "nudge" the expert into the wrong correction?

- To address reviewer comments about the case studies, we have shortened the geologic descriptions (removing sub-discipline jargon) and improved the connection between the case study section and the preceding bias review section. Details of these changes are discussed in comments below.
- Regarding the identification of biases by AI – this is something we cover in the conclusion when discussing nudge design and the importance of "explainability".

In terms of style, the first 3 sections are a pleasure to read, but include some repetitions, e.g., many statements are made in Section 1 and then repeated in more detail in later sections. So I suggest to look for redundant statements and shorten those sections a bit.

- Redundancies were checked for and removed, specifically repetitious references to IT/AI solutions to debiasing.

For Section 4 my main suggestions would be to 1) work hard on clarity in the case studies of how exactly they connect to the biases discussed earlier; 2) spell out the way forward, i.e. how could the lessons learned here be generalized to other applications. In fact, I might even suggest to drastically shorten + de-emphasize the case studies to be only 1-2 paragraphs each, and maybe moving the rest to an appendix, then to focus on the main message of the paper in terms of fighting general biases and how to do that instead. I don't know, however, whether the remaining material warrants publication in this venue.

- To address reviewer comments about the case studies, we have shortened the geologic descriptions (removing sub-discipline jargon) and improved the connection between the case study section and the preceding bias review section.
- For case study 1, we have improved the connection to the preceding bias review section by explicitly detailing how automated flights can address susceptibility to anchoring bias in field decisions about where to fly. The title of case study 1 was changed to: "Optimizing field data collection with UAVs to minimize anchoring bias". We have also removed many extra details about UAVs not directly pertinent to the case study. Most of the changes were to the first three paragraphs
- For case study 2, we have improved the connection to preceding bias review section by explicitly detailing how seismic interpretation aids (built into software) can address

susceptibility to availability bias during interpretation. The title of case study 2 was changed to: "Fault interpretations in 3D seismic image data to minimize availability bias". We also removed details about the technique of automated horizon tracking (including the accompanying Figure 5) which are not directly pertinent to the case study. Most of the changes were to the first three paragraphs

P. 5, Line 16-17: Mentions the three types of biases for the 3rd time. Too much repetition.
- This is only the second time the bias types are mentioned (besides the abstract). We feel the repetition in this instance is warranted.
- No changes to manuscript

P. 5, Line 30. There's something missing here. "Over ??? have found ... "
- Formatting error, missing "⅔"
- Changed to "two-thirds"

P. 11. There is a Section 4.1.1, but no 4.1.2. P. 14,
- We intend section 4.1.1 to be included as an appendix or be contained within some box that distinguishes it from the main text.
- No changes to manuscript

Line 24. "of of"
- Noted.
- Changed to "of"

**How can geologic decision making under uncertainty be improved?**

Cristina G. Wilson[1,2]
Clare E. Bond[3]
Thomas F. Shipley[1]

[revised manuscript text omitted]